# OUTLIER-ROBUST OPTIMAL TRANSPORT

## ABSTRACT

Optimal transport (OT) provides a way of measuring distances between distributions that depends on the geometry of the sample space. In light of recent advances in solving the OT problem, OT distances are widely used as loss functions in minimum distance estimation. Despite its prevalence and advantages, however, OT is extremely sensitive to outliers. A single adversarially-picked outlier can increase OT distance arbitrarily. To address this issue, in this work we propose an outlier-robust OT formulation. Our formulation is convex but challenging to scale at a first glance. We proceed by deriving an *equivalent* formulation based on cost truncation that is easy to incorporate into modern stochastic algorithms for regularized OT. We demonstrate our model applied to mean estimation under the Huber contamination model in simulation as well as outlier detection on real data.

## 1 INTRODUCTION

Optimal transport is a fundamental problem in applied mathematics. In its original form (Monge, 1781), the problem entails finding the minimum cost way to transport mass from a prescribed probability distribution $\mu$ on $\mathcal{X}$ to another prescribed distribution $\nu$ on $\mathcal{X}$. Kantorovich (1942) relaxed Monge's formulation of the optimal transport problem to obtain the Kantorovich formulation:

$$\text{OT}(\mu, \nu) \triangleq \min_{\Pi \in \mathcal{F}(\mu,\nu)} \mathbb{E}_{(X_1, X_2) \sim \Pi}\big[c(X_1, X_2)\big], \tag{1.1}$$

where $\mathcal{F}(\mu, \nu)$ is the set of couplings between $\mu$ and $\nu$ (probability distributions on $\mathcal{X} \times \mathcal{X}$ whose marginals are $\mu$ and $\nu$) and $c$ is a cost function, where we typically assume $c(x, y) \geq 0$ and $c(x, x) = 0$. Compared to other notions of distance between probability distributions, optimal transport uniquely depends on the geometry of the sample space.

Recent advancements in optimization for optimal transport (Cuturi, 2013; Solomon et al., 2015; Genevay et al., 2016; Seguy et al., 2018) enabled its broad adaptation in machine learning applications where geometry of the data is important. See (Peyré & Cuturi, 2018) for a survey. Optimal transport has found applications in natural language processing (Kusner et al., 2015; Huang et al., 2016; Alvarez-Melis & Jaakkola, 2018; Yurochkin et al., 2019), generative modeling (Arjovsky et al., 2017), clustering (Ho et al., 2017), domain adaptation (Courty et al., 2014; 2017), large-scale Bayesian modeling (Srivastava et al., 2018), and many other domains.

Many applications use OT as a loss in an optimization problem of the form:

$$\theta \in \arg\min_{\theta \in \Theta} \text{OT}(\mu_n, \nu_\theta), \tag{1.2}$$

where $\{\nu_\theta\}_{\theta \in \Theta}$ is a collection of parametric models, $\mu_n$ is the empirical distribution of the samples. Such estimators are called *minimum Kantorovich estimators (MKE)* (Bassetti et al., 2006). They are popular alternatives to likelihood-based estimators, especially in generative modeling. For example, when $\text{OT}(\cdot, \cdot)$ is the Wasserstein-1 distance and $\nu_\theta$ is a generator parameterized by a neural network with weights $\theta$, equation 1.2 corresponds to the Wasserstein GAN (Arjovsky et al., 2017).

One drawback of optimal transport is its sensitivity to outliers. Because *all* the mass in $\mu$ must be transported to $\nu$, a small fraction of outliers can have an outsized impact on the optimal transport problem. For statistics and machine learning applications in which the data is corrupted or noisy, this is a major issue. For example, the poor performance of Wasserstein GANs in the presence of outliers was noted in the recent works on outlier-robust generative learning with $f$-divergence GANs (Chao et al., 2018; Wu et al., 2020). The problem of outlier-robustness in MKE has not been studied, with the exception of two concurrent works (Staerman et al., 2020; Balaji et al., 2020).

In this paper, we propose a modification of OT to address its sensitivity to outliers. Our formulation can be used as a loss in equation 1.2 so that it is robust to a small fraction of outliers in the data. To keep things simple, we consider the $\epsilon$-contamination model (Huber & Ronchetti, 2009). Let $\nu_{\theta_0}$ be a member of a parametric model $\{\nu_\theta : \theta \in \Theta\}$ and let

$$\mu = (1 - \epsilon)\nu_{\theta_0} + \epsilon\tilde{\nu},$$

where $\mu$ is the data-generating distribution, $\epsilon > 0$ is the fraction of outliers, and $\tilde{\nu}$ is the distribution of the outliers. Although the fraction of outliers is capped at $\epsilon$, the value of the outliers is arbitrary, so the outliers may have an arbitrarily large impact on the optimal transport problem. Our goal is to modify the optimal transport problem so that it is more robust to outliers. We have in mind the downstream application of learning $\theta_0$ from (samples from) $\mu$ in the $\epsilon$-contamination model. Our main contributions are as follows:

1. We propose a robust OT formulation that is suitable for statistical estimation in the $\epsilon$-contamination model using MKE.
2. We show that our formulation is equivalent to the original OT problem with a clipped transport cost. This connection enables us to leverage the voluminous literature on computational optimal transport to develop efficient algorithm to perform MKE robust to outliers.
3. Our formulation enables a new application of optimal transport: outlier detection in data.

## 2 PROBLEM FORMULATION

### 2.1 ROBUST OT FOR MKE

To promote outlier-robustness in MKE, we need to allow the corresponding OT problem to ignore the outliers in the data distribution $\mu$. The $\epsilon$-contamination model imposes a cap on the fraction of outliers, so it is not hard to see that $\|\mu - \nu_{\theta_0}\|_{\mathsf{TV}} \leq \epsilon$, where $\|\cdot\|_{\mathsf{TV}}$ is the total-variation norm defined as $\|\mu\|_{\mathsf{TV}} = \int \frac{1}{2}|\mu(\mathrm{d}x)|$. This suggests we solve a TV-constrained/regularized version of equation 1.2. The constrained version

$$\min_{\theta \in \Theta, \tilde{\mu}} \quad \mathrm{OT}(\tilde{\mu}, \nu_\theta)$$
$$\text{subject to} \quad \|\mu - \tilde{\mu}\|_{\mathsf{TV}} \leq \epsilon$$

suffers from identification issues. In particular, it cannot distinguish between "clean" distributions within TV distance $\epsilon$ of $\nu_{\theta_0}$. This makes it unsuitable as a loss function for statistical estimation, because it cannot lead to a consistent estimator. However, its regularized counterpart

$$\min_{\theta \in \Theta, s} \mathrm{OT}(\mu + s, \nu_\theta) + \lambda\|s\|_{\mathsf{TV}}, \tag{2.1}$$

where $\lambda > 0$ is a regularization parameter, does not suffer from this issue. In the rest of this paper, we work with the TV-regularized formulation equation 2.1.

The main idea of our formulation is to allow for modifications of $\mu$, while penalizing their magnitude and ensuring that the modified $\mu$ is still a probability measure. Below we formulate this intuition in an optimization problem titled ROBOT (ROBust Optimal Transport):

**Formulation 1:**

$$\mathrm{ROBOT}(\mu, \nu) = \begin{cases} \min_{\substack{\Pi \in \mathcal{F}^+(\mathbb{R}^d \times \mathbb{R}^d) \\ s \in \mathcal{F}(\mathbb{R}^d)}} & \int C(x, y)\, \Pi(\mathrm{d}x, \mathrm{d}y) + \lambda\|s\|_{\mathsf{TV}} \\[2ex] \text{subject to} & \int_{B \times \mathbb{R}^d} \Pi(\mathrm{d}x, \mathrm{d}y) = \int_B (\mu(\mathrm{d}x) + s(\mathrm{d}x)) \geq 0 \\ & \qquad \forall\, B \in \mathscr{B}(\mathbb{R}^d)\ (\text{Borel } \sigma\text{-algebra}) \\[1ex] & \int_{\mathbb{R}^d \times C} \Pi(\mathrm{d}x, \mathrm{d}y) = \int_C \nu(\mathrm{d}y)\ \forall\, C \in \mathscr{B}(\mathbb{R}^d) \\[1ex] & \int s(\mathrm{d}x) = 0. \end{cases} \tag{2.2}$$

Here $\mathcal{F}(\mathbb{R}^d)$ denotes the set of all signed measures with finite total variation on $\mathbb{R}^d$, $\mathcal{F}^+(\mathbb{R}^d \times \mathbb{R}^d)$ is the set of all measures with finite total variation on $\mathbb{R}^d \times \mathbb{R}^d$.

The first and the last constraints ensure that $\mu + s$ is a valid probability measure, while $\lambda \|s\|_{\text{TV}}$ penalizes the amount of modifications in $\mu$. It is worth noting that we can identify exact locations of outliers in $\mu$ by inspecting $\mu + s$, i.e. if $\mu(x) + s(x) = 0$, then $x$ got eliminated and is an outlier.

ROBOT, unlike classical OT, guarantees that an adversarially picked outliers can not increase the distance arbitrarily. Let $\tilde{\mu} = (1 - \epsilon)\mu + \epsilon\mu_c$, i.e. $\tilde{\mu}$ is $\mu$ contaminated with outliers from $\mu_c$, and let $\nu$ be an arbitrary measure (in MKE, $\tilde{\mu}$ is the contaminated data and $\nu$ is the model we learn). Adversary can arbitrarily increase $\text{OT}(\tilde{\mu}, \nu)$ by manipulating the outlier distribution $\mu_c$. For ROBOT we have the following bound:

**Theorem 2.1.** *Let $\tilde{\mu} = (1 - \epsilon)\mu + \epsilon\mu_c$ for some $\epsilon \in [0, 1)$, then*

$$\text{ROBOT}(\tilde{\mu}, \nu) \leq (\text{OT}(\mu, \nu) + \lambda\epsilon\|\mu - \mu_c\|_{\text{TV}}) \wedge \lambda\|\tilde{\mu} - \nu\|_{\text{TV}} \wedge \text{OT}(\tilde{\mu}, \nu). \quad (2.3)$$

This bound has two key takeaways: since TV norm of any two distributions is bounded by 1, adversary can not increase $\text{ROBOT}(\tilde{\mu}, \nu)$ arbitrarily; in the absence of outliers, ROBOT is bounded by classical OT. See Appendix C for the proof.

**Related work** We note connection between equation 2.2 and unbalanced OT (UOT) (Chizat., 2017; Chizat et al., 2018). UOT is typically formulated by replacing TV norm with $\text{KL}(\mu + s | \mu)$ and adding an analogous term for $\nu$. Chizat et al. (2018) studied entropy regularized UOT with various divergences penalizing marginal violations. Optimization problems similar to equation 2.2 have also been considered outside of the ML literature (Piccoli & Rossi, 2014; Liero et al., 2018). We are unaware of prior applications of UOT to outlier-robustness, but it was studied in the concurrent work of Balaji et al. (2020). Another relevant variation of OT is partial OT (Figalli, 2010; Caffarelli & McCann, 2010). It may also be considered for outlier-robustness, but it has a drawback of forcing mass destruction rather than adjusting marginals to ignore outliers when they are present. A concurrent work by Staerman et al. (2020) took a different path: they replaced the expectation in the Wasserstein-1 dual with a median-of-means to promote robustness. It is unclear what is the corresponding primal, making it hard to interpret as an optimal transport problem.

A major challenge with the aforementioned methods, including our Formulation 1, is the difficulty of the optimization problem. This is especially the case for MKEs, where a transport problem has to be solved in every iteration to obtain the gradient of the model parameters. Chizat et al. (2018) proposed a Sinkhorn-like algorithm for entropy regularized UOT, but it is not amenable to stochastic optimization. Balaji et al. (2020) proposed a stochastic optimization algorithm based on the UOT dual, but it requires two additional neural networks (total of four including dual potentials) to parameterize modified marginal distributions (i.e., $\mu + s$ and analogous one for $\nu$). Optimizing with a median-of-means in the objective function as in (Staerman et al., 2020) is also challenging. The key contribution of our work is a formulation *equivalent* to equation 2.2, which is *easily compatible* with the large body of classical OT optimization techniques (Cuturi, 2013; Solomon et al., 2015; Genevay et al., 2016; Seguy et al., 2018).

**More efficient equivalent formulation** At a first glance, there are two issues with equation 2.2: it appears asymmetric and it is unclear if it can be optimized efficiently. Below we present an *equivalent* formulation that is free of these issues:

**Formulation 2:**

$$\text{ROBOT}(\mu, \nu) = \begin{cases} \min_{\Pi \in \mathcal{F}^+(\mathbb{R}^d \times \mathbb{R}^d)} & \int C_\lambda(x, y)\Pi(\mathrm{d}x, \mathrm{d}y) \\ \text{subject to} & \int_{B \times \mathbb{R}^d} \Pi(\mathrm{d}x, \mathrm{d}y) = \int_B \mu(\mathrm{d}x) \ \forall \, B \in \mathscr{B}(\mathbb{R}^d) \\ & \int_{\mathbb{R}^d \times C} \Pi(\mathrm{d}x, \mathrm{d}y) = \int_C \nu(\mathrm{d}y) \ \forall \, C \in \mathscr{B}(\mathbb{R}^d), \end{cases} \quad (2.4)$$

where $C_\lambda$ is the *truncated cost* function defined as $C_\lambda(x, y) = C(x, y) \wedge 2\lambda$. Looking at equation 2.4, it is not apparent that it adds robustness to MKE, but it is symmetric, easy to combine

with entropic regularization by simply truncating the cost, and benefits from stochastic optimization algorithms (Genevay et al., 2016; Seguy et al., 2018). This formulation also has a distant relation to the idea of loss truncation for achieving robustness (Shen & Sanghavi, 2019). Pele & Werman (2009) considered the Earth Mover Distance (discrete OT) with truncated cost to achieve computational improvements; they also mentioned its potential to promote robustness against outlier noise but did not explore this direction.

In Section 3, we establish *equivalence* between the two ROBOT formulations, equation 2.2 and equation 2.4. This equivalence allows us to obtain an efficient algorithm based on equation 2.4 for robust MKE. We also provide a simple procedure for computing optimal $s$ in equation 2.2 from the solution of equation 2.4, enabling a new OT application: outlier detection. We verify the effectiveness of robust MKE and outlier detection in our experiments in Section 4. Before presenting the equivalence proof, we formulate the discrete analogs of the two ROBOT formulations for their practical value.

## 2.2 DISCRETE ROBOT FORMULATIONS

In practice we typically encounter samples from the distributions, rather then the distributions themselves. Sampling is also built into stochastic optimization. In this subsection, we present the discrete versions of the ROBOT formulations. The key detail is that, in equation 2.2, $\mu, \nu$ and $s$ are all supported on $\mathbb{R}^d$, while in the discrete case the empirical measures $\mu_n \in \Delta^{n-1}$ and $\nu_m \in \Delta^{m-1}$ are supported on a set of points ($\Delta^r$ is the unit probability simplex in $\mathbb{R}^r$). As a result, to formulate a discrete version of equation 2.2, we need to augment $\mu_n$ and $\nu_m$ with each others' supports. To be precise, let $\text{supp}(\mu_n) = \{X_1, \ldots, X_n\}$ and $\text{supp}(\nu_m) = \{Y_1, \ldots, Y_m\}$. Define $\mathcal{C} = \{Z_1, Z_2, \ldots, Z_{m+n}\} = \{X_1, \ldots, X_n, Y_1, \ldots, Y_m\}$. Then discrete analog of equation 2.2 is

**Formulation 1 (discrete):**

$$\text{ROBOT}(\mu_n, \nu_m) = \begin{cases} \min_{\substack{\Pi \in \mathbb{R}^{(m+n) \times (m+n)} \\ \mathbf{s} \in \mathbb{R}^{m+n}}} & \langle C_{aug}, \Pi \rangle + \lambda \left[ \|s_1\|_1 + \|t_1\|_1 \right] \\ \text{subject to} & \Pi 1_{m+n} = \begin{bmatrix} \mu_n + s_1 \\ t_1 \end{bmatrix}, \quad \Pi^\top 1_{m+n} = \begin{bmatrix} 0 \\ \nu_m \end{bmatrix} \\ \Pi \succeq 0, \quad 1_{m+n}^\top \mathbf{s} = 0, \end{cases} \tag{2.5}$$

where $C_{aug} \in \mathbb{R}^{(m+n) \times (m+n)}$ is the augmented cost function $C_{aug,i,j} = c(Z_i, Z_j)$ ($c$ is the ground cost, e.g., squared Euclidean distance), $\mathbf{s} = (s_1, t_1)$ and $1_r$ is the vector all ones in $\mathbb{R}^r$. The TV norm got replaced with its discrete analog, the $L_1$ norm. Similarly to its continuous counterpart, the optimization problem is harder than the typical OT due to additional constraint optimization variable $\mathbf{s}$ and increased cost matrix size.

The discrete analog of equation 2.4 is straightforward:

**Formulation 2 (discrete):**

$$\text{ROBOT}(\mu_n, \nu_m) = \begin{cases} \min_{\Pi \in \mathbb{R}^{n \times m}} & \langle C_\lambda, \Pi \rangle \\ \text{subject to} & \Pi 1_n = \mu_n, \quad \Pi^\top 1_m = \nu_m, \quad \Pi \succeq 0, \end{cases} \tag{2.6}$$

where $C_{\lambda,i,j} = c(X_i, Y_j) \wedge 2\lambda$. As in the continuous case, it is easy to adapt modern (regularized) OT solvers without any computational overhead. As in the continuous case, formulations of equation 2.5 and equation 2.6 are equivalent. It is also possible to recover $\mathbf{s}$ of equation 2.5 from the solution of equation 2.6 to perform outlier detection.

**Two-sided formulation** So far we have assumed that one of the input distributions does not have outliers, which is the setting of MKE, where the clean distribution corresponds to the model we learn. In some applications, both distributions may be corrupted. To address this case, we provide an *equivalent* two-sided formulation, analogous to UOT with TV norm:

**Formulation 3 (two-sided):**

$$\text{ROBOT}(\mu_n, \nu_m) = \begin{cases} \min\limits_{\substack{\Pi \in \mathbb{R}^{(m+n)\times(m+n)} \\ \mathbf{s}_1 \in \mathbb{R}^{m+n}, \mathbf{s}_2 \in \mathbb{R}^{m+n}}} & \langle C_{aug}, \Pi \rangle + \lambda \left[ \|s_1\|_1 + \|t_1\|_1 + \|s_2\|_1 + \|t_2\|_1 \right] \\ \text{subject to} & \Pi 1_{m+n} = \begin{bmatrix} \mu_n + s_1 \\ t_1 \end{bmatrix}, \quad \Pi^\top 1_{m+n} = \begin{bmatrix} s_2 \\ \nu_m + t_2 \end{bmatrix} \\ & \Pi \succeq 0, \quad 1_{m+n}^\top \mathbf{s}_1 = 0, \quad 1_{m+n}^\top \mathbf{s}_2 = 0. \end{cases}$$
(2.7)

where $\mathbf{s}_1 = (s_1^\top, t_1^\top)^\top$ and $\mathbf{s}_2 = (s_2^\top, t_2^\top)^\top$.

# 3 EQUIVALENCE OF THE ROBOT FORMULATIONS

In this section we present our main theorem, which demonstrates the equivalence between two formulations of the robust optimal transport:

**Theorem 3.1.** *For any two measures $\mu$ and $\nu$, ROBOT$(\mu, \nu)$ has same value for both the formulations, i.e., Formulation 1 is equivalent to Formulation 2 both for continuous and discrete case. Moreover, we can recover optimal coupling of one formulation from the other.*

Below we sketch the proof of this theorem and highlight some important techniques used in the proof. We focus on the discrete case as it is more intuitive and has concrete practical implications in our experiments. A complete proof can be found in Appendix A. Please also see Appendix A.2 for the proof of equivalence between Formulations 1, 2 and 3 in the discrete case.

## 3.1 PROOF SKETCH

In the remainder of this section we consider the discrete case, i.e., equation 2.5 for Formulation 1 (F1) and equation 2.6 for Formulation 2 (F2). Suppose $\Pi_2^*$ is an optimal solution of F2. Then we construct a feasible solution $\Pi_1^*, \mathbf{s}_1^* = (s_1^*, t_1^*)$ of F1 based on $\Pi_2^*$ with the same value of the objective function as F2 and claim that $(\Pi_1^*, \mathbf{s}_1^*)$ is an optimal solution. We prove the claim by contradiction: if $(\Pi_1^*, \mathbf{s}_1^*)$ is not optimal, then there exists another pair $(\tilde{\Pi}_1, \tilde{\mathbf{s}}_1)$ which is optimal for F1 with strictly less objective value. We then construct another feasible solution $\Pi_{2,new}^*$ of Formulation 2 which has the same objective value as of $(\tilde{\Pi}_1, \tilde{\mathbf{s}}_1)$ for F1. This implies $\Pi_{2,new}^*$ has strictly less objective value for F2 than $\Pi_2^*$, which is a contradiction.

The two main pillars of this proof are (1) to construct a feasible solution of F1 starting from a feasible solution of F2 and (2) to show that the solution constructed is indeed optimal for F1. Hence step (1) gives a recipe to construct an optimal solution of F1 starting from an optimal solution of F2. We elaborate the first point in the next subsection, which has practical implications for outlier detection. The other point is more technical; interested readers may go through the proof in Appendix A.1.

---

**Algorithm 1** Generating optimal solution of F1 from F2

---

1: Start with $\Pi_2^* \in \mathbb{R}^{n \times m}$, an optimal solution of Formulation 2.
2: Create an augmented matrix $\Pi \in \mathbb{R}^{m+n \times m+n}$ with all 0. Divide $\Pi$ into four blocks:

$$\Pi = \begin{bmatrix} \underbrace{\Pi_{11}}_{n \times n} & \underbrace{\Pi_{12}}_{n \times m} \\ \underbrace{\Pi_{21}}_{m \times n} & \underbrace{\Pi_{22}}_{m \times m} \end{bmatrix}$$

3: Set $\Pi_{12} \leftarrow \Pi_2^*$ and collect all the indices $\mathcal{I} = \{(i,j) : C_{i,j} > 2\lambda\}$.
4: Set $\Pi_{12}(i,j) \leftarrow 0$ for $(i,j) \in \mathcal{I}$.
5: Set $\Pi_{22}(j,j) \leftarrow \sum_{i=1}^n \Pi_2^*(i,j) \mathbb{1}_{(i,j) \in \mathcal{I}}$ for all $1 \le j \le m$ and set $\Pi_1^* \leftarrow \Pi$.
6: Set $s_1^*(i) \le \sum_{j=1}^m \Pi_2^*(i,j) \mathbb{1}_{(i,j) \in \mathcal{I}}$ for all $1 \le i \le n$.
7: Set $t_1^*(j) = \Pi_{22}(j,j)$ for all $1 \le j \le m$.
8: return $\Pi_1^*, s_1^*, t_1^*$.

---

## 3.2 GOING FROM FORMULATION 2 TO FORMULATION 1

Let $\Pi_2^*$ (respectively $\Pi_1^*$) be an optimal solution of F2 (respectively F1). Recall that $\Pi_1^*$ has dimension $(m + n) \times (m + n)$. From the column sum constraint in F1, we need to take the first $n$ columns of $\Pi_1^*$ to be exactly 0, whereas the last $m$ columns must sum up to $\nu_m$. For any matrix $A$, we denote by $A[(a : b) \times (c : d)]$ the submatrix consisting of rows from $a$ to $b$ and columns from $c$ to $d$. Our main idea is to put a modified version of $\Pi_2^*$ in $\Pi_1^*[(1 : n) \times (n + 1 : m + n)]$ and make $\Pi_1^*[(n+1 : m+n) \times (n+1 : m+n)]$ diagonal. First we describe how to modify $\Pi_2^*$. Observe that, if for some $(i, j)$ $C_{i,j} > 2\lambda$, we expect $X_i \in \text{supp}(\mu_n)$ to be an outlier resulting in high transportation cost, which is why we truncate the cost in F2. Therefore, to get an optimal solution of F1, we make the corresponding value of optimal plan 0 and dump the mass into the corresponding slack variable $t_1^*$ in the diagonal of the bottom right submatrix. This changes the row sum, which is taken care of by $s_1^*$. But, as we are not moving this mass outside the corresponding column, the column sum of $\Pi_1^*[(1 : (m + n)) : ((n + 1) : (m + n))]$ remains same as column sum of $\Pi_2^*$, which is $\nu_n$. We summarize this procedure in Algorithm 1.

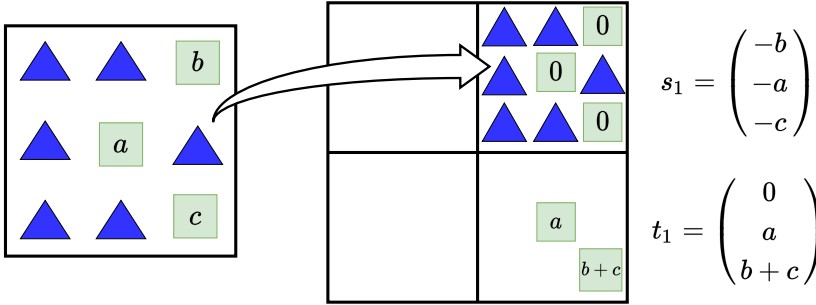

Figure 1: Constructing optimal solution of Formulation 1 from optimal solution of Formulation 2.

**Example.** In Figure 1, we provide an example to visualize the construction. On the left, we have $\Pi_2^*$, an optimal solution of Formulation 2. The blue triangles denote the positions where the corresponding cost value is $\leq 2\lambda$, and light-green squares denote the positions where the corresponding value of the cost matrix is $> 2\lambda$. To construct an optimal solution $\Pi_1^*$ of Formulation 1 from this $\Pi_2^*$, we first create an augmented matrix of size $6 \times 6$. We keep all the entries of of left $6 \times 3$ sub-matrix as 0 (in this picture blank elements indicate 0). On the right submatrix, we put $\Pi_2^*$ into the top-right block, but remove the masses from light-green squares, i.e. where cost value is $> 2\lambda$, and put it in the diagonal entries of the bottom right block as shown in Figure 1. This mass contributes to the slack variables $s_1$ and $t_1$, and this augmented matrix along with $s_1, t_1$ give us an optimal solution of Formulation 1.

## 3.3 OUTLIER DETECTION WITH ROBOT

Our construction algorithm has practical consequences for outlier detection. Suppose we have two datasets, a clean dataset $\nu_m$ (i.e., has no outliers) and an outlier-contaminated dataset $\mu_n$. We can detect the outliers in $\mu_n$ without directly solving costly Formulation 1 by following Algorithm 2. In this algorithm, $\lambda$ is a regularization parameter that can be chosen via cross-validation or heuristically (see Section 4.2 for an example). In Section 4.2, we use this algorithm to perform outlier detection on image data.

---

**Algorithm 2** Outlier detection in contaminated data

---

1: Start with $\mu_n$ (contaminted data) and $\nu_m$ (clean data).
2: Solve Formulation 2 and obtain $\Pi_2^*$ using a suitable value of $\lambda$.
3: Use Algorithm 1 to obtain $\Pi_1^*, s_1^*, t_1^*$ from $\Pi_2^*$.
4: Find $\mathcal{I}$, the set of all the indices where $\mu_n + s_1^* = 0$.
5: Return $\mathcal{I}$ as the indices of outliers in $\mu_n$.

---

Table 1: Robust mean estimation with GANs using different distribution divergences. True mean is $\eta_0 = \mathbf{0}_5$; sample size $n = 1000$; contamination proportion $\epsilon = 0.2$. We report results over 30 experiment restarts.

| Contamination | JS Loss | SH Loss | RKL Loss | ROBOT | UOT |
|---|---|---|---|---|---|
| $\mathcal{N}(0.1 \cdot \mathbf{1_5}, I_5)$ | $\mathbf{0.09} \pm 0.03$ | $0.11 \pm 0.03$ | $0.115 \pm 0.03$ | $0.1 \pm 0.03$ | $0.1 \pm 0.04$ |
| $\mathcal{N}(0.5 \cdot \mathbf{1_5}, I_5)$ | $0.23 \pm 0.04$ | $0.24 \pm 0.05$ | $0.24 \pm 0.05$ | $\mathbf{0.117} \pm 0.03$ | $0.2 \pm 0.04$ |
| $\mathcal{N}(1 \cdot \mathbf{1_5}, I_5)$ | $0.43 \pm 0.05$ | $0.43 \pm 0.06$ | $0.43 \pm 0.06$ | $0.261 \pm 0.06$ | $\mathbf{0.25} \pm 0.05$ |
| $\mathcal{N}(2 \cdot \mathbf{1_5}, I_5)$ | $0.67 \pm 0.07$ | $0.67 \pm 0.08$ | $0.67 \pm 0.08$ | $0.106 \pm 0.03$ | $\mathbf{0.1} \pm 0.03$ |

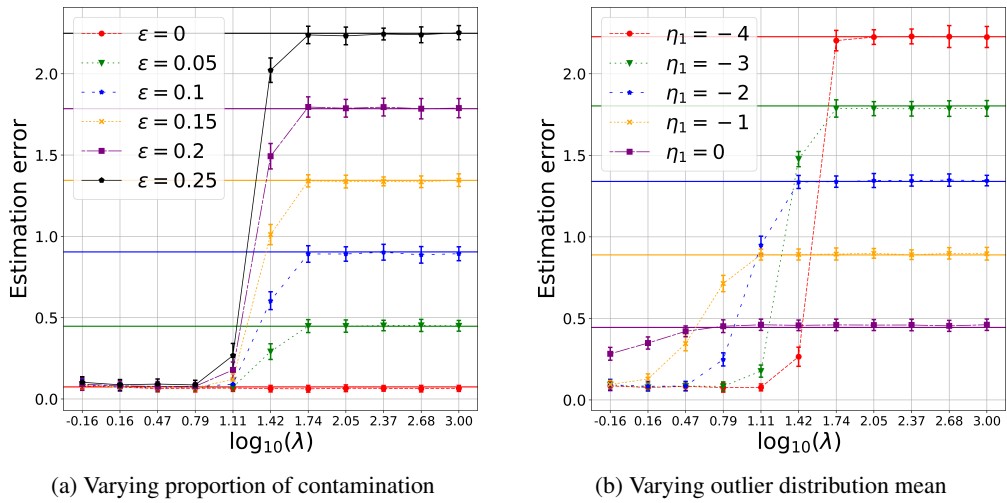

(a) Varying proportion of contamination

(b) Varying outlier distribution mean

Figure 2: Empirical study of regularization hyperparameter $\lambda$ sensitivity

## 4 EMPIRICAL STUDIES

To evaluate effectiveness of ROBOT, we consider the task of robust mean estimation under the Huber contamination model. The data is generated from $(1 - \epsilon)\mathcal{N}(\eta_0, I_d) + \epsilon\mathcal{N}(\eta_1, I_d)$ and the goal is to estimate $\eta_0$. Prior work has advocated for using $f$-divergence GANs (Chao et al., 2018; Wu et al., 2020) for this problem and pointed out inefficiencies of Wasserstein GAN in the presence of outliers. We show that our robust OT formulation allows us to estimate the uncontaminated mean $\eta_0$ comparably or better than a variety of $f$-divergence GANs. We also use this simulated setup to study sensitivity to the regularization hyperparameter $\lambda$.

In our second experiment, we present a new application of optimal transport enabled by ROBOT. Suppose we have collected a curated dataset $\nu_m$ (i.e., we know that it has no outliers)—such data collection is expensive, and we want to benefit from it to automate subsequent data collection. Let $\mu_n$ be a second dataset collected "in the wild," i.e., it may or may not have outliers. We demonstrate how ROBOT can be used to identify outliers in $\mu_n$ using the curated dataset $\nu_m$.

### 4.1 ROBUST MEAN ESTIMATION

Following Wu et al. (2020), we consider a simple generator of the form $g_\theta(x) = x + \theta$, $x \sim \mathcal{N}(0, I_d)$, $d$ is the data dimension. The basic idea of robust mean estimation with GANs is to minimize various distributional divergences between samples from $g_\theta$ and observed data simulated from $(1 - \epsilon)\mathcal{N}(\eta_0, I_d) + \epsilon\mathcal{N}(\eta_1, I_d)$. The goal is to estimate $\eta_0$ with $\theta$. To efficiently implement ROBOT GAN, we use a standard min-max optimization approach: solve the inner max (ROBOT) and use gradient descent for the outer min parameter. To solve ROBOT, it is straightforward to adopt any of the prior stochastic regularized OT solvers: the only modification is the truncation of the cost entries as in equation 2.6. We use the stochastic algorithm for semi-discrete regularized OT from

(Genevay et al., 2016, Algorithm 2). We summarize ROBOT GAN in Algorithm 3. Line 5 - Line 10 perform the inner optimization where we solve entropy regularized OT dual with truncated cost and Line 11 - Line 12 perform gradient update of $\theta$.

---

**Algorithm 3** ROBOT GAN

---

1: **Input:** robustness regularizion $\lambda$, entropic regularization $\alpha$, data distribution $\mu_n \in \Delta^{n-1}$, $supp(\mu_n) = \mathcal{X} = [X_1, \ldots, X_n]$, steps sizes $\tau$ and $\gamma$
2: **Initialize:** Initialize $\theta = \theta_{init}$, set number of iterations $M$ and $L$, $i = 0$, $\mathbf{v} = \tilde{\mathbf{v}} = \mathbf{0}$.
3: **for** $j = 1, \ldots, M$ **do**
4:    Generate $\tilde{z} \sim \mathcal{N}(0, I_d)$ and set $z = \tilde{z} + \theta$.
5:    Set the cost vector $\mathbf{c} \in \mathbb{R}^n$ as $\mathbf{c}(k) = c(X_k, z) \wedge 2\lambda$ for $k = 1, \ldots, n$.
6:    **for** $i = 1, \ldots, L$ **do**                               $\triangleright$ solve entropy regularized OT dual
7:       Set $\mathbf{h} \leftarrow \frac{\tilde{\mathbf{v}} - \mathbf{c}}{\alpha}$ and do the normalized exponential transformation $\mathbf{u} \leftarrow \frac{e^{\mathbf{h}}}{\langle \mathbf{1}, e^{\mathbf{h}} \rangle}$.
8:       Calculate the gradient $\nabla \tilde{\mathbf{v}} \leftarrow \mu_n - \mathbf{u}$.
9:       Update $\tilde{\mathbf{v}} \leftarrow \tilde{\mathbf{v}} + \gamma \nabla \tilde{\mathbf{v}}$ and $\mathbf{v} \leftarrow (1/(j+i))\tilde{\mathbf{v}} + (j + i - 1/(j+i))\mathbf{v}$.
10:    Do the same transformation of $\mathbf{v}$ as in Step 7, i.e. set $\mathbf{h} \leftarrow \frac{\mathbf{v} - \mathbf{c}}{\alpha}$ and set $\Pi \leftarrow \frac{e^{\mathbf{h}}}{\langle \mathbf{1}, e^{\mathbf{h}} \rangle}$.
11:    Set $\Pi(k) = 0$ for $k$ such that $C(X_k, z) > 2\lambda$ for $k = 1, \ldots, n$.
12:    Calculate gradient with respect to $\theta$ as $\nabla \theta = 2 \left[ z \sum_k \Pi(k) - \mathcal{X}^\top \Pi \right]$
13:    Update $\theta \leftarrow \theta - \tau \nabla \theta$.
14: **Ouput:** $\theta$

---

For the $f$-divergence GANs (Nowozin et al., 2016) we use the code of Wu et al. (2020) for GANs with Jensen-Shannon (JS) loss, squared Hellinger (SH) loss and Reverse Kullback-Leibler (RKL) loss. For the exact expression of these divergences see Table 1 of Wu et al. (2020). We report estimation error measured by the Euclidean distance between true uncontaminated mean $\eta_0$ and estimated mean $\theta$ for various contamination distributions in Table 1. ROBOT GAN performs well across all considered contamination distributions. As the difference between true mean $\eta_0$ and contamination mean $\eta_1$ increases, the estimation error of all methods tends to increase. However, when it becomes easier to distinguish outliers from clean samples, i.e., $\eta_1 = 2 \cdot \mathbf{1_5}$, performance of ROBOT noticeably improves.

We also compared to the Sinkhorn-based UOT algorithm (Chizat et al., 2018) available in the Python Optimal Transport (POT) library (Flamary & Courty, 2017); to obtain a UOT GAN, we modified steps 5-11 of Algorithm 3 for computing $\Pi$. Unsurprisingly, both ROBOT and UOT perform similarly: recall equivalence to Formulation 3, which is similar to UOT with TV norm. The key insight of our work is the equivalence to classical OT with truncated cost, that greatly simplifies optimization and allows to use existing stochastic OT algorithms. In this experiment, the sample size $n = 1000$ is sufficiently small for the Sinkhorn-based UOT POT implementation to be effective, but it breaks in the experiment we present in Section 4.2. We also tried the code of Balaji et al. (2020) based on CVXPY (Diamond & Boyd, 2016), but it is too slow even for the $n = 1000$ sample size.

In the previous experiment, we set $\lambda = 0.5$. Now we demonstrate empirically that there is a broad range of $\lambda$ values performing well. In Figure 2a, we study sensitivity of $\lambda$ under various contamination proportions $\epsilon$ holding $\eta_0 = \mathbf{1}_5$ and $\eta_1 = 5 \cdot \mathbf{1}_5$ fixed. Horizontal lines correspond to $\lambda = \infty$, i.e., vanilla OT. The key observations are: there is a wide range of $\lambda$ efficient at all contamination proportions, and ROBOT is always at least as good as vanilla OT (even when there is no contamination $\epsilon = 0$). In Figure 2b, we present a similar study varying the mean of the contamination distribution and holding $\epsilon = 0.2$ fixed. We see that as the contamination distribution gets closer to the true distribution, it becomes harder to pick a good $\lambda$, but the performance is always at least as good as the vanilla OT (horizontal lines).

## 4.2 OUTLIER DETECTION FOR DATA COLLECTION

Our robust OT formulation equation 2.5 enables outlier identification. Let $\nu_m$ be a clean dataset and $\mu_n$ potentially contaminated with outliers. Recall that ROBOT allows modification of one of the input distributions to eliminate potential outliers. We can identify outliers in $\mu_n$ as follows: if $\mu_n(i) + s_1^*(i) = 0$, then $X_i$, the $i$th point in $\mu_n$, is an outlier. Instead of directly solving equation 2.5,

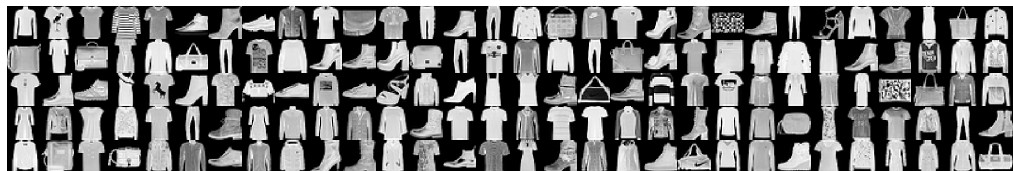

Figure 3: Random sample of outliers detected by ROBOT from a dataset of MNIST digits contaminated with Fashion MNIST images.

which may be inefficient, we use our equivalence results and solve an easier optimization problem equation 2.6, followed by recovering $\mathbf{s}$ to find outliers via Algorithm 2.

Let $\nu_m$ be a clean dataset consisting of 10k MNIST digits and $\mu_n$ be a dataset collected "in the wild" consisting of (different) 8k MNIST digits and 2k Fashion MNIST images. We compute ROBOT$(\mu_n, \nu_m)$ to identify outlier Fashion MNIST images in $\mu_n$. For each point in $\mu_n$ we obtain a prediction, outlier or clean, which allows us to evaluate accuracy. ROBOT outlier detection is 90% accurate in this experiment. We also comment on $\lambda$ selection: since we know that $\nu_m$ is clean, we can subsample two datasets from it, compute vanilla OT to obtain transportation plan $\Pi$ and set $\lambda$ to be half the maximum distance between matched elements, i.e. $2\lambda = \max_{i,j}\{C_{ij} : \Pi_{ij} > 0\}$, where $C$ is the cost matrix for the two subsampled datasets. This procedure is essentially estimating maximum distance between matched clean samples. We also present a random sample of outliers identified by our method in Figure 3. All of the sampled outliers are Fashion MNIST images, although 90% accuracy suggests that some of the outliers were not identified. Decreasing $\lambda$ can help to find more outliers, but may result in some clean samples being mistaken for outliers. We conclude that ROBOT can be used to assist in data collection once an initial set of clean data has been acquired. As we mentioned previously, the Sinkhorn-based UOT POT implementation is too expensive for this experiment due to larger sample size, yielding memory errors on a personal laptop with 16GB RAM.

For comparison, we also consider a heuristic distance-based approach for identifying outliers. We estimate diameter $\tau$ of the set of clean dataset $\nu_m$ by taking the 99th percentile of the pairwise distance matrix of samples in $\nu_m$. If outliers and clean data have disjoint support, we can adopt a simple heuristic: for each sample in the potentially contaminated $\mu_n$ compute an average distance to the clean samples in $\nu_m$ and declare a sample as an outlier if this average distance is greater than the diameter $\tau$ of the clean data. The accuracy of this procedure is 85.4%, inferior to the ROBOT accuracy of 90%. The disjoint support assumption justifying the distance-based heuristic might be too strong in practice. ROBOT continues to be effective even when the supports of clean and outlier distributions are not easily separable.

## 5  SUMMARY AND DISCUSSION

We proposed and studied ROBOT, a robust formulation of optimal transport. We showed that although the problem is seemingly asymmetric and challenging to optimize, there is an equivalent formulation based on cost truncation that is symmetric and compatible with modern stochastic optimization methods for OT.

ROBOT closely resembles unbalanced optimal transport (UOT). In our formulation, we added a TV regularizer to the vanilla optimal transport problem. This is motivated by the $\epsilon$-contamination model. In UOT, the TV regularizer is typically replaced with a KL divergence. Other choices of the regularizer may lead to new properties and applications. Studying equivalent, simpler formulations of UOT with different divergences may be a fruitful future work direction.

From the practical perspective, in our experiments we observed no degradation of ROBOT GAN in comparison to OT GAN, even when there were no outliers. It is possible that replacing OT with ROBOT may be beneficial for various machine learning applications of OT. Data encountered in practice may not be explicitly contaminated with outliers, but it often has errors and other deficiencies, suggesting that a "no-harm" robustness is desirable.

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

# A  PROOF OF THEOREM 3.1

## A.1  PROOF OF DISCRETE VERSION

*Proof.* Define a matrix $\Pi$ as:

$$\Pi(i,j) = \begin{cases} 0, & \text{if } C(i,j) > 2\lambda \\ \Pi_2^*(i,j), & \text{otherwise} \end{cases}$$

Also define $s \in \mathbb{R}^n$ and $t \in \mathbb{R}^m$ as:

$$s_1^*(i) = -\sum_{j=1}^m \Pi_2^*(i,j)\mathbb{1}_{C(i,j)>2\lambda}$$

and similarly define:

$$t_1^*(j) = \sum_{i=1}^n \Pi_2^*(i,j)\mathbb{1}_{C(i,j)>2\lambda}$$

These vectors corresponds to the row sums and the column sums of the elements of the optimal transport plan of Formulation 2, where the cost function exceeds $2\lambda$. Note that, these co-ordinates of the optimal transport plan corresponding to those co-ordinates of cost matrix, where the cost is greater than $2\lambda$ and contribute to the objective value via their sum only, hence any different arrangement of these transition probabilities with same sum gives the same objective value.

Now based on this $\Pi$ obtained we construct a feasible solution of Formulation 1 following Algorithm 1:

$$\Pi_1^* = \begin{bmatrix} \mathbf{0} & \Pi \\ \mathbf{0} & \text{diag}(t_1^*) \end{bmatrix}$$

The row sums of $\Pi_1^*$ is:

$$\Pi_1^* \mathbf{1} = \begin{bmatrix} \mu_n + s_1^* \\ t_1^* \end{bmatrix}$$

and it is immediate from the construction that the column sums of $\Pi_1^*$ is $\nu_m$. Also as:

$$\sum_{i=1}^n s_1^*(i) = \sum_{j=1}^m t_1^*(j) = \sum_{(i,j):C_{i,j}>2\lambda} \Pi_2^*(i,j)$$

and $s_1^* \preceq 0, t_1^* \succeq 0$, we have:

$$\mathbf{1}^\top (\mu_n + s_1^* + t_1^*) = \mathbf{1}^\top p = 1 \,.$$

Therefore, we have $(\Pi_1^*, s_1^*, t_1^*)$ is a feasible solution of Formulation 1. Now suppose this is not an optimal solution. Pick an optimal solution $\tilde{\Pi}, \tilde{s}, \tilde{t}$ of Formulation 1 so that:

$$\langle C_{aug}, \tilde{\Pi} \rangle + \lambda \left[ \|\tilde{s}\|_1 + \|\tilde{t}\|_1 \right] < \langle C_{aug}, \Pi_1^* \rangle + \lambda \left[ \|s_1^*\|_1 + \|t_1^*\|_1 \right]$$

The following two lemmas provide some structural properties of any optimal solution of Formulation 1:

**Lemma A.1.** *Suppose $\Pi_1^*, s_1^*, t_1^*$ are optimal solution for Formulation 1. Divide $\Pi_1^*$ into four parts corresponding to augmentation as in algorithm 1:*

$$\Pi_1^* = \begin{bmatrix} \Pi_{1,11}^* & \Pi_{1,12}^* \\ \Pi_{1,21}^* & \Pi_{1,22}^* \end{bmatrix}$$

*Then we have $\Pi_{1,11}^* = \Pi_{1,21}^* = \mathbf{0}$ and $\Pi_{1,22}^*$ is a diagonal matrix.*

**Lemma A.2.** *If $\Pi_1^*, s_1^*, t_1^*$ is an optimal solution of Formulation 1 then:*

*1. If $C_{i,j} > 2\lambda$ then $\Pi_1^*(i,j) = 0$.*
*2. If $C_{i,j} < 2\lambda$ for some $i$ and for all $1 \leq j \leq n$, then $s_1^*(i) = 0$.*
*3. If $C_{i,j} < 2\lambda$ for some $j$ and for all $1 \leq i \leq m$, then $t_1^*(j) = 0$.*
*4. If $C_{i,j} < 2\lambda$ then $s_1^*(i)t_1^*(j) = 0$.*

We provide the proofs in the next subsection. By Lemma A.1 we can assume without loss of generality:

$$\tilde{\Pi} = \begin{bmatrix} \mathbf{0} & \tilde{\Pi}_{12} \\ \mathbf{0} & \text{diag}(\tilde{t}) \end{bmatrix}$$

Now based on $\left( \tilde{\Pi}, \tilde{s}, \tilde{t} \right)$ we create a feasible solution namely $\Pi^*_{2,new}$ of Formulation 2 as follows: Define the set of indices $\{i_1, \cdots, i_k\}$ and $\{j_1, \ldots, j_l\}$ as:

$$\tilde{s}_{i_1}, \tilde{s}_{i_2}, \ldots, \tilde{s}_{i_k} > 0 \quad \text{and} \quad \tilde{t}_{j_1}, \tilde{t}_{j_2}, \ldots, \tilde{t}_{j_l} > 0.$$

Then by part (4) of Lemma A.2 we have $C_{i_\alpha, j_\beta} > 2\lambda$ for $\alpha \in \{1, \ldots, k\}$ and $\beta \in \{1, \ldots, l\}$. Also by part (2) of Lemma A.2 the value of transport plan at these co-ordinates is 0. Now distribute the mass of slack variables in these co-ordinates such that the marginals of new transport plan becomes exactly $\mu_n$ and $\nu_m$. This new transport plan is our $\Pi^*_{2,new}$. Recall that, $\|\tilde{s}\|_1 = \|\tilde{t}\|_1$. Hence, here the regularizer value decreases by $2\lambda\|\tilde{s}\|_1$ and the cost value increased by exactly $2\lambda\|\tilde{s}\|_1$ as we are truncating the cost. Hence we have:

$$\begin{aligned} \langle C_\lambda, \Pi^*_{2,new} \rangle &= \langle C_{aug}, \tilde{\Pi} \rangle + \lambda \left[ \|\tilde{s}\|_1 + \|\tilde{t}\|_1 \right] \\ &< \langle C_{aug}, \Pi^*_1 \rangle + \lambda \left[ \|s^*_1\|_1 + \|t^*_1\|_1 \right] \\ &= \langle C_\lambda, \Pi^*_2 \rangle \end{aligned}$$

which is contradiction as $\Pi^*_2$ is the optimal solution of Formulation 2. This completes the proof for the discrete part.

$\square$

## A.2  PROOF OF EQUIVALENCE FOR TWO SIDED FORMULATION

Here we prove that our two sided formulation, i.e. Formulation 3 (equation 2.7) is equivalent to Formulation 1 (equation 2.5) for the discrete case. Towards that end, we introduce another auxiliary formulation and show that both Formulation 1 and Formulation 3 are equivalent to the following auxiliary formulation of the problem.

**Formulation 4:**

$$W_{\mathbf{R},\mathbf{L},4}(p,q) = \begin{cases} \min_{\Pi \in \mathbb{R}^{m \times n}, s_1 \in \mathbb{R}^m, s_2 \in \mathbb{R}^n} & \langle C, \Pi \rangle + \lambda \left[ \|s_1\|_1 + \|s_2\|_1 \right] \\ \text{subject to} & \Pi \mathbf{1}_n = p + s_1 \\ & \Pi^T \mathbf{1}_m = q + s_2 \\ & \Pi \succeq 0 \end{cases} \tag{A.1}$$

First we show that Formulation 1 and Formulation 4 are equivalent in a sense that they have the same optimal objective value.

**Theorem A.3.** *Suppose $C$ is a cost function such that $C(x, x) = 0$. Then Formulation 1 and Formulation 4 has same optimal objective value.*

*Proof.* Towards that end, we show that given one optimal variables of one formulation we can get optimal variables of other formulation with the same objective value. Before going into details we need the following lemma whose proof is provided in Appendix B:

**Lemma A.4.** *Suppose $\Pi^*_4, s^*_{4,1}, s^*_{4,2}$ are the optimal variables of Formulation 4. Then $s^*_{4,1} \preceq 0$ and $s^*_{4,2} \preceq 0$.*

Now we prove that optimal value of Formulation 1 and Formulation 4 are same. Let $(\Pi^*_1, s^*_{1,1}, t^*_{1,1})$ is an optimal solution of Formulation 1. Then we claim that $(\Pi^*_1, s^*_{1,1}, t^*_{1,1})$ is also an optimal solution of Formulation 4. Clearly it is feasible solution of Formulation 4. Suppose it is not optimal, i.e. there exists another optimal solution $(\tilde{\Pi}_4, \tilde{s}_{4,1}, \tilde{s}_{4,2})$ such that:

$$\langle C, \tilde{\Pi}_4 \rangle + \lambda(\|\tilde{s}_{4,1}\|_1 + \|\tilde{s}_{4,2}\|_2) < \langle C, \Pi^*_{1,12} \rangle + \lambda(\|s^*_{1,1}\|_1 + \|t^*_{1,1}\|_1)$$

Now based on $(\tilde{\Pi}_4, \tilde{s}_{4,1}, \tilde{s}_{4,2})$ we construct a feasible solution of Formulation 1 as follows:

$$\tilde{\Pi}_1 = \begin{bmatrix} \mathbf{0} & \tilde{\Pi}_4 \\ \mathbf{0} & -\text{diag}(\tilde{s}_{4,2}) \end{bmatrix}$$

Note that we proved in Lemma A.4 $\tilde{s}_{4,2} \preceq 0$, hence we have $\tilde{\Pi}_1 \succeq 0$. Now as the column sums of $\tilde{\Pi}_4$ is $q + \tilde{s}_{4,2}$, we have column sums of $\tilde{\Pi}_1 = [\mathbf{0} \ q^\top]^\top$ and the row sums are $[(p + \tilde{s}_{4,1})^\top \ \tilde{s}_{4,2}^\top]^\top$. Hence we take $\tilde{s}_{1,1} = \tilde{s}_{4,1}$ and $\tilde{s}_{1,2} = \tilde{s}_{4,2}$. Then it follows:

$$\begin{aligned} \langle C_{aug}, \tilde{\Pi}_1 \rangle + \lambda \left[ \|\tilde{s}_{1,1}\|_1 + \|\tilde{s}_{1,2}\|_1 \right] &= \langle C, \tilde{\Pi}_4 \rangle + \lambda \left[ \|\tilde{s}_{4,1}\|_1 + \|\tilde{s}_{4,2}\|_1 \right] \\ &< \langle C, \Pi_{1,12}^* \rangle + \lambda \left[ \|s_{1,1}^*\|_1 + \|t_{1,1}^*\|_1 \right] \\ &= \langle C_{aug}, \Pi_1^* \rangle + \lambda \left[ \|s_{1,1}^*\|_1 + \|t_{1,1}^*\|_1 \right] \end{aligned}$$

This is contradiction as we assumed $(\Pi_1^*, s_{1,1}^*, t_{1,2}^*)$ is an optimal solution of Formulation 1. Therefore we conclude $(\Pi_1^*, s_{1,1}^*, t_{1,1}^*)$ is also an optimal solution of Formulation 4 which further concludes Formulation 1 and Formulation 4 have same optimal values. This completes the proof of the theorem. □

**Theorem A.5.** *The optimal objective value of Formulation 3 and Formulation 4 are same.*

*Proof.* Like in the proof of Theorem A.3 we also prove couple of lemmas.

**Lemma A.6.** *Any optimal transport plan $\Pi_3^*$ of Formulation 3 has the following structure: If we write,*

$$\Pi_3^* = \begin{bmatrix} \Pi_{3,11}^* & \Pi_{3,12}^* \\ \Pi_{3,21}^* & \Pi_{3,22}^* \end{bmatrix}$$

*then $\Pi_{3,11}^*$ and $\Pi_{3,22}^*$ are diagonal matrices and $\Pi_{3,21}^* = \mathbf{0}$.*

**Lemma A.7.** *If $s_{3,1}^*, t_{3,1}^*, s_{3,2}^*, t_{3,2}^*$ are four optimal slack variables in Formulation 3, then $s_{3,1}^*, t_{3,1}^* \preceq 0$ and $s_{3,2}^*, t_{3,2}^* \succeq 0$.*

*Proof.* The line of argument is same as in proof of Lemma A.4. □

Next we establish equivalence. Suppose $(\Pi_3^*, s_{3,1}^*, t_{3,1}^*, s_{3,2}^*, t_{3,2}^*)$ are optimal values of Formulation 3. We claim that $(\Pi_{3,12}^*, s_{3,1}^* - s_{3,2}^*, t_{3,1}^* - t_{3,2}^*)$ forms an optimal solution of Formulation 4. The objective value will then also be same as $s_{3,1}^* \preceq 0, s_{3,2}^* \succeq 0$ (Lemma A.7) implies $\|s_{3,1}^* - s_{3,2}^*\|_1 = \|s_{3,1}^*\|_1 + \|s_{3,2}^*\|_1$ and similarly $t_{3,1}^* \preceq 0, t_{3,2}^* \succeq 0$ implies $\|t_{3,1}^* - t_{3,2}^*\|_1 = \|t_{3,1}^*\|_1 + \|t_{3,2}^*\|_1$. Feasibility is immediate. Now for optimality, we again prove by contradiction. Suppose they are not optimal. Then lets say $\tilde{\Pi}_4, \tilde{s}_{4,1}, \tilde{s}_{4,2}$ are an optimal triplet of Formulation 4. Now construct another feasible solution of Formulation 3 as follows: Set $\tilde{s}_{3,2} = \tilde{t}_{3,2} = 0, \tilde{s}_{3,1} = \tilde{s}_{4,1}$ and $\tilde{t}_{3,1} = \tilde{s}_{4,2}$. Set the matrix as:

$$\tilde{\Pi}_3 = \begin{bmatrix} \mathbf{0} & \tilde{\Pi}_4 \\ \mathbf{0} & -\text{diag}(\tilde{s}_{4,2}) \end{bmatrix}$$

Then it follows that $\left( \tilde{\Pi}_3, \tilde{s}_{3,1}, \tilde{s}_{3,2}, \tilde{t}_{3,1}, \tilde{t}_{3,2} \right)$ is a feasible solution of Formulation 3. Finally we have:

$$\begin{aligned} &\langle C_{aug}, \tilde{\Pi}_3 \rangle + \lambda \left[ \|\tilde{s}_{3,1}\|_1 + \|\tilde{s}_{3,2}\|_1 + \|\tilde{t}_{3,1}\|_1 + \|\tilde{t}_{3,2}\|_1 \right] \\ &= \langle C_{aug}, \tilde{\Pi}_3 \rangle + \lambda \left[ \|\tilde{s}_{4,1}\|_1 + \|\tilde{s}_{4,2}\|_1 \right] \\ &= \langle C, \tilde{\Pi}_4 \rangle + \lambda \left[ \|\tilde{s}_{4,1}\|_1 + \|\tilde{s}_{4,2}\|_1 \right] \\ &< \langle C, \Pi_{3,12}^* \rangle + \lambda \left[ \|s_{3,1}^* - s_{3,2}^*\|_1 + \|t_{3,1}^* - t_{3,2}^*\|_1 \right] \\ &= \langle C_{aug}, \Pi_3^* \rangle + \lambda \left[ \|s_{3,1}^*\|_1 + \|s_{3,2}^*\|_1 + \|t_{3,1}^*\|_1 + \|t_{3,2}^*\|_1 \right] \end{aligned}$$

This contradicts the optimality of $(\Pi_3^*, s_{3,1}^*, s_{3,2}^*, t_{3,1}^*, t_{3,2}^*)$. This completes the proof. □

### A.3 Proof of continuous version

*Proof.* In this proof we denote by $F_1$ the optimization problem of equation equation 2.2 and by $F_2$ the optimization problem equation equation 2.4. Assume that $\mu_n$ and $\nu_m$ denote the respective empirical measures relative to $\mu, \nu$. From Villani (2009), we know that $\mu_n, \nu_n$ converge weakly to $\mu$ and $\nu$ respectively. Therefore, $ROBOT_2(\mu_n, \mu) \to 0$. Similary for $\nu_n$ and $\nu$. Thus, by triangle inequality,

$$\lim_{n \to \infty} |F_2(\mu_n, \nu_n) - F_2(\mu, \nu)| = 0.$$

But $ROBOT_2(\mu_n, \nu_n) = ROBOT_1(\mu_n, \nu_n)$. Therefore, our proof is complete if we can show that

$$\lim_{n,m \to \infty} |F_1(\mu_n, \nu_m) - F_1(\mu, \nu)| \to 0.$$

Let $\mathscr{S} = \{s \text{ signed measure } : \mu + s \text{ is a probability measure in } \mathbb{R}^d\}$. For $s \in \mathscr{S}$, define

$$W(\mu+S,\nu) = \begin{cases} \min_{\Pi \in \mathcal{F}(\mathbb{R}^d \times \mathbb{R}^d)} & \int C(x,y)\, \Pi(\mathrm{d}x, \mathrm{d}y) + \lambda \|s\|_{TV} \\[2mm] \text{subject to} & \int_A \Pi(\mathrm{d}x, \mathrm{d}y) \geq 0 \; \forall\, A \in \mathscr{B}(\mathbb{R}^d \times \mathbb{R}^d) \\[2mm] \mu(\mathrm{d}x) + s(\mathrm{d}x)) \geq 0 \\[2mm] & \forall\, B \in \mathscr{B}(\mathbb{R}^d) \\[2mm] & \int_{\mathbb{R}^d \times C} \Pi(\mathrm{d}x, \mathrm{d}y) = \int_C \nu(\mathrm{d}y) \;\; \forall\, C \in \mathscr{B}(\mathbb{R}^d) \end{cases}$$

By Lemma A.8, $\exists\, s \in \mathscr{S}$ such that $ROBOT(\mu, \nu) = W(\mu + s, \nu) + \lambda \|s\|_{TV}$. Let $s = s^+ - s^-$, where $s^+$ and $s^-$ are positive measures on $\mathbb{R}^d$. Let $\|s\|_{TV} = \gamma$. Then, $\|s^-\|_{TV} = \|s^+\|_{TV} = \gamma/2$.

Then consider $X_1, \ldots, X_n \sim (P - s^-)/(1 - \gamma)$, $Y_1, \ldots, Y_n \sim s^-/\gamma$, $Z_1, \ldots, Z_n \sim s^+/\gamma$. Then for any bounded continuous function $f$,

$$\lim_{n \to \infty} \sum_i f(X_i)/n = \int f(x) \frac{(P - s^-)}{(1 - \gamma)}(\mathrm{d}x)$$

$$\lim_{n \to \infty} \sum_i f(Z_i)/n = \int f(x) \frac{s^+}{\gamma}(\mathrm{d}x) \tag{A.2}$$

Therefore, the distribution given by $(P + s)_n = \dfrac{(1 - \gamma)}{n} \sum_i \delta_{X_i} + \dfrac{\gamma}{n} \sum_i \delta_{Z_i}$ satisfies, $(P + s)_n \xrightarrow{\mathscr{L}} P + s$, and therefore from (Villani, 2009), $\lim_{n \to \infty} W_C((P + S)_n, \nu_n) \to W(P + S, Q)$. Here $\delta_x$ is the Dirac mass at $x$. Moreover, $\|s_n\| = \|s\|$, where $s_n$ satisfies $s_n = \dfrac{\gamma}{n}(\sum_i \delta_{Z_i} - \sum_i \delta_{Y_i})$.

Also, $ROBOT(\mu_n, \nu_n) \leq W((P + s)_n, \nu_n) + \lambda \|s\|_{TV}$, and therefore $ROBOT_2(\mu, \nu)) = \limsup_{n \to \infty} ROBOT(\mu_n, \nu_n) \leq ROBOT(\mu, \nu)$.

Now, let $\tilde{s}_n$ satisfy $W_1(\mu_n + \tilde{s}_n, \nu_n) + \lambda \|\tilde{s}_n\|_{TV} = ROBOT(\mu_n, \nu_n)$. Such an $\tilde{s}_n$ exists by the proof of the discrete part because $\mu_n, \nu_n$ are discrete measures.

Then, similar to the Step 1 in the proof of Lemma A.8, there exists a probability measure $\mu \oplus s$ and a subsequence $\{n_k\}_{k \geq 1}$ such that $\mu_{n_k} + s_{n_k}$ almost surely converges weakly to $\mu \oplus s$.

Moreover, similar to Step 2 of Lemma A.8 $W_1(\mu_{n_k} + s_{n_k}, \mu \oplus s) \to 0$ as well as $\|s_{n_k}\|_{TV} \to \|\mu \oplus s - \mu\|_{TV}$. Thus, $W_1(\mu_{n_k} + s_{n_k}, \nu_{n_k}) + \lambda \|s_{n_k}\|_{TV} \to W_1(\mu \oplus s, \nu) + \lambda \|\mu \oplus s - \mu\|_{TV}$. But by the proof of the discrete part $ROBOT(\mu_{n_k}, \nu_{n_k}) = ROBOT_2(\mu_{n_k}, \nu_{n_k}) \to ROBOT_2(\mu, \nu)$. Therefore, with $s = \mu \oplus s - \mu$, $W_1(\mu + s, \nu) + \lambda \|s\|_{TV} = ROBOT_2(\mu, \nu)$.

Therefore, $ROBOT_2(\mu, \nu) = \limsup_{n \to \infty} ROBOT(\mu_n, \nu_n) \geq ROBOT(\mu, \nu)$. Thus the equality holds. $\qquad\square$

**Lemma A.8.** *Assume that $\mu, \nu$ is such that $\int \|x\| \mathrm{d}\mu, \int \|x\| \mathrm{d}\nu < \infty$. Moreover, assume that $C(x, y)$ in equation 2.2 is the $l_1$ norm, i.e., $C(x, y) = \|x - y\|$. Then, there exists $s$ with $\mu + s$ being a probability measure such that*

$$W_1(\mu + s, \nu) + \lambda \|s\|_{TV} = ROBOT(\mu, \nu), \tag{A.3}$$

*where $W_1$ is the Wasserstein-1 norm with the cost function $C(\cdot, \cdot)$ as mentioned above.*

*Proof.* Let $\mu_n, \nu_m$ be the empirical measures relative to $\mu, \nu$ respectively. We know that since $\mu_n, \nu_m$ are discrete, there exists $s_n$ satisfying $W_1(\mu_n + s_n, \nu_m) = ROBOT(\mu_n, \nu_m)$. We provide the proof in the following steps.

**Step 1:** Almost surely $\mu \times \nu$, there exists a subsequence $\{n_k\}_{k \geq 1}$ such that $\{\mu_n + s_n\}_n$ and $\{\nu_n\}_n$ is relatively compact.

$\mu$ and $\nu$ are probability measures on $\mathbb{R}^d$ and are therefore tight.
Let $K_\epsilon$ be such that $P_\mu(X \notin K_\epsilon), P_\nu(Y \notin K_\epsilon) \leq \epsilon/4$.

Consider the empirical distributions $\nu_n = \sum_i \delta_{Y_i}/n, \mu_n \sum_i \delta_{X_i}/n$ of $\nu, \mu$ respectively. Here, $X_i \sim \mu$ and $Y_i \sim \nu$.

Fix an $\omega$. Then $\{X_1, \ldots, X_n, Y_1, \ldots, Y_n\}$ is fixed. Now by the construction for the discrete case, $s_n$ has support in $\{X_1, \ldots, X_n, Y_1, \ldots, Y_n\}$.

Let $T_n$ be the optimal transport map from $\mu_n$ to $\nu_n$. Then, for every $i \leq n$, there exists a unique $j \leq n$, such that $T_n(X_i) = Y_j$. Define $\tau_n : \{1, \ldots, n\} \to \{1, \ldots, n\}$ such that $\tau_n(i) = j$ if $T_n(X_i) = Y_j$. Then $\mu_n + s_n = \sum_i \delta_{Z_i}/n$, where $Z_i = X_i$ or $Y_{\tau_n(i)}$ and $\delta_x$ is the Dirac delta mass at $x$.

Then, let $Z \sim \mu_n + s_n$

$$P_\omega(Z \notin K_\epsilon | \mu_n + s_n) \leq \sum_i \mathbb{1}_{(X_i \notin K_\epsilon)}/n + \sum_i \mathbb{1}_{(Y_i \notin K_\epsilon)}/n \tag{A.4}$$

Therefore, $\mathbb{E}(P_\omega(Z \notin K_\epsilon | \mu_n + s_n)) \leq \epsilon/2$. Moreover, $Var(P_\omega(Z \notin K_\epsilon | \mu_n + s_n)) = o(n^{-1}) \to 0$. Therefore, $\lim_{n \to \infty} P_{\mu^n \times \nu^n}(P_\omega(Z \notin K_\epsilon | \mu_n + s_n) \leq \epsilon) \to 1$. Therefore $\mu_n + s_n$ is almost surely tight and thus by Prokhorov's Theorem also relatively compact.

**Step 2:** Therefore, for $\omega$ almost surely, there exists a subsequence $\{n_k\}_{k \geq 1}$ such that $\mu_{n_k} + s_{n_k}$ converges weakly to a limit (dependent on $\omega$) $\mu \oplus s$ which is a probability measure. Moreover, $\int \|x\| \mathrm{d}(\mu_{n_k} + s_{n_k}) < \infty$ almost surely. By Bolzano-Weierstrass Theorem, there exists a further subsequence $\{n_{k_l}\}_l$ such that $\int \|x\| \mathrm{d}(\mu_{n_{k_l}} + s_{n_{k_l}}) \to \int \|x\| \mathrm{d}(\mu + s)$ almost surely. For the sake of convenience, without loss of generality, we will replace the sub-subsequence $\{n_{k_l}\}_l$ with $\{n_k\}_{k \geq 1}$ henceforth.

Thus, by Theorem 6.9 of (Villani, 2009), $W_1(\mu_{n_k} + s_{n_k}, \mu \oplus s) \to 0$ almost surely. Moreover, $W_1(\mu_{n_k}, \mu) \to 0$ almost surely. Therefore $\|s_{n_k}\|_{TV} \to \|\mu \oplus s - \mu\|_{TV}$ almost surely.

**Step 3:** Consider an arbitrary $S = S^+ - S^-$, such that $S^+$ and $S^-$ are positive measures on $\mathbb{R}^d$, and $\mu + S$ is a probability measure. Let $\|S\|_{TV} = \gamma$. Then, $\|S^-\|_{TV} = \|S^+\|_{TV} = \gamma/2$.

Then consider $X_1, \ldots, X_n \sim (\mu - S^-)/(1 - \gamma), Y_1, \ldots, Y_n \sim S^-/\gamma, Z_1, \ldots, Z_n \sim S^+/\gamma$. Then for any bounded continuous function $f$,

$$\lim_{n \to \infty} \sum_i f(X_i)/n = \int f(x) \frac{(P - s^-)}{(1 - \gamma)} (\mathrm{d}x)$$

$$\lim_{n \to \infty} \sum_i f(Z_i)/n = \int f(x) \frac{s^+}{\gamma} (\mathrm{d}x) \tag{A.5}$$

Therefore, the distribution given by $(\mu + S)_n(A) = (1 - \gamma) \sum_i \mathbb{1}_{X_i \in A} + (\gamma) \sum_i \mathbb{1}_{Z_i \in A}$ satisfies, $(\mu + S)_n \xrightarrow{\mathscr{L}} \mu + S$, and therefore from (Villani, 2009), $\lim_{n \to \infty} W_1((\mu + S)_n, \nu_n) \to W_1(\mu + S, \nu)$. Moreover, $\|S_n\|_{TV} = \|S\|_{TV}$, where $S_n$ satisfies $S_n(A) = \frac{\gamma}{n} \sum_i \mathbb{1}_{Z_i \in A} - \sum_i \mathbb{1}_{Y_i \in A}$.

But, $W_1(\mu_{n_k} + s_{n_k}, \nu_{n_k}) + \lambda\|s_{n_k}\|_{TV} \leq W_1((\mu + S)_{n_k}, \nu_{n_k}) + \lambda\|S_{n_k}\|_{TV}$. Therefore, taking limits, $W_1(\mu \oplus s, \nu) + \lambda\|\mu \oplus s - \mu\|_{TV} \leq W_1(\mu + S, \nu) + \lambda\|S\|_{TV}$, and thus the proof holds with $s = \mu \oplus s - \mu$.

$\square$

# B    PROOF OF ADDITIONAL LEMMAS

## B.1    PROOF OF LEMMA A.1

*Proof.* The fact that $\Pi^*_{1,11} = \Pi^*_{1,21} = \mathbf{0}$ follows from the fact that $\Pi^*_1 \succeq 0$ and $\Pi^*_1 \mathbf{1} = \mathbf{Q}$. To prove that $\Pi^*_{1,22}$ is diagonal, we use the fact that the any diagonal entry the cost matrix is 0. Now suppose $\Pi^*_{1,22}$ is not diagonal. Then define a matrix $\hat{\Pi}$ as following: set $\hat{\Pi}_{11} = \hat{\Pi}_{21} = \mathbf{0}$, $\hat{\Pi}_{12} = \Pi^*_{1,12}$ and:

$$\hat{\Pi}_{22}(i,j) = \begin{cases} \sum_{k=1}^{m} \Pi^*_{1,22}(k,i), & \text{if } j = i \\ 0, & \text{if } j \neq i \end{cases}$$

Also define $\hat{s} = s^*_1$ and $\hat{t}$ as $\hat{t}(i) = \hat{\Pi}_{22}(i,i)$. Then clearly $(\hat{\Pi}, \hat{s}, \hat{t})$ is a feasible solution of Formulation 1. Note that:

$$\|\hat{t}\|_1 = \mathbf{1}^\top \hat{\Pi}_{22} \mathbf{1} = \mathbf{1}^\top \Pi^*_{1,22} \mathbf{1} = \|t^*_1\|_1$$

and by our construction $\langle C_{aug}, \hat{\Pi} \rangle < \langle C_{aug}, \Pi^*_1 \rangle$. Hence $(\hat{\Pi}, \hat{s}, \hat{t})$ reduces the value of the objective function of Formulation 1 which is a contradiction. This completes the proof. $\square$

## B.2    PROOF OF LEMMA A.2

*Proof.* 1. Suppose $\Pi^*_1(i,j) > 0$. Then dump this mass to $s^*_1(j)$ and make it 0. In this way $\langle C_{aug}, \Pi^*_1 \rangle$ will decrease by $> 2\lambda \Pi^*_1(i,j)$ and the regularizer value will increase by atmost $2\lambda \Pi^*_1(i,j)$, resulting in overall reduction in the objective value, which leads to a contradiction.
2. Suppose each entry of $i^{th}$ row of $C$ is $< 2\lambda$. Then if $s^*_1(i) > 0$, we can distribute this mass in the $i^{th}$ row such that, $s^*_1(i) = a_1 + a_2 + \cdots + a_m$ with the condition that $t^*_1(j) \geq a_j$. Now we reduce $t^*_1$ as:

$$t^*_1(j) \leftarrow t^*_1(j) - a_j$$

Hence the value $\langle C_{aug}, \Pi^*_1(i,j) \rangle$ will increase by a value $< 2\lambda s^*_1(i)$ but the value of regularizer will decrease by the value of $2\lambda s^*_1(i)$, resulting in overall decrease in the value of objective function.
3. Same as proof of part (2) by interchanging row and column in the argument.
4. Suppose not. Then choose $\epsilon < s^*_1(i) \wedge t^*_1(j)$, Add $\epsilon$ to $\Pi^*_1(i,j)$. Hence the cost function value $\langle C_{aug}, \Pi^*_1 \rangle$ will increase by $< 2\lambda\epsilon$ but the regularizer value will decrease by $2\lambda\epsilon$, resulting in overall decrease in the objective function.

$\square$

## B.3    PROOF OF LEMMA A.4

*Proof.* For the notational simplicity, we drop the subscript 4 now as we will only deal with the solution of Formulation 4 and there will be no ambiguity. We prove the Lemma by contradiction. Suppose $s^*_{1,i} > 0$. Then we show one can come up with another solution $(\tilde{\Pi}, \tilde{s}_1, \tilde{s}_2)$ of Formulation 4 such that it has lower objective value. To construct this new solution, make:

$$\tilde{s}_{1,j} = \begin{cases} s^*_{1,j}, & \text{if } j \neq i \\ 0, & \text{if } j = i \end{cases}$$

Now to change the optimal transport plan, we will only change $i^{th}$ row of $\Pi^*$. We subtract $a_1, a_2, \ldots, a_n \geq 0$ from $i^{th}$ column of $\Pi^*$ in such a way, such that none of the elements are negative. Hence the column sum will be change, i.e. the value of $\tilde{s}_2$ will be:

$$\tilde{s}_{2,j} = s^*_{2,j} - a_j \quad \forall 1 \leq j \leq n.$$

Now clearly from our construction:

$$\langle C, \tilde{\Pi} \rangle \leq \langle C, \Pi^* \rangle$$

For the regularization part, note that, as we only reduced $i^{th}$ element of $s_1^*$, we have $\|\tilde{s}_1\|_1 = \|s_1^*\|_1 - s_{1,i}^*$. And by simple triangle inequality,

$$\|\tilde{s}_2\|_1 \leq \|s_2^*\|_1 + \|a_1\|_1 = \|s_2^*\|_1 + s_{1,i}^*$$

by construction $a_i$'s, as $a_i \geq 0$ and $\sum_i a_i = s_{1,i}^*$. Hence we have:

$$\|\tilde{s}_1\|_1 + \|\tilde{s}_2\|_1 \leq \|s_1^*\|_1 - s_{1,i}^* + \|s_2^*\|_1 + s_{1,i}^* = \|s_1^*\|_1 + \|s_2^*\|_1.$$

Hence the value corresponding to regularizer will also decrease. This completes the proof. □

### B.4 Proof of Lemma A.6

*Proof.* We prove this lemma by contradiction. Suppose $\Pi_3^*$ does not have the structure mentioned in the statement of Lemma. Construct another transport plan for Formulation 3 $\tilde{\Pi}_3$ as follows: Keep $\tilde{\Pi}_{3,12} = \Pi_{3,12}^*$ and set $\tilde{\Pi}_{3,12} = \mathbf{0}$. Construct the other parts as:

$$\tilde{\Pi}_{3,11}(i,j) = \begin{cases} \sum_{k=1}^m \Pi_{3,11}^*(i,k) + \sum_{k=1}^n \Pi_{3,21}^*(k,i), & \text{if } i = j \\ 0, & \text{if } i \neq j \end{cases}$$

and

$$\tilde{\Pi}_{3,22}(i,j) = \begin{cases} \sum_{k=1}^n \Pi_{3,22}^*(k,i), & \text{if } i = j \\ 0, & \text{if } i \neq j \end{cases}$$

It is immediate from the construction that:

$$\langle C_{aug}, \tilde{\Pi}_3 \rangle \leq \langle C_{aug}, \Pi_3^* \rangle$$

As for the regularization term: Note the by our construction $\tilde{s}_4$ will be same as $s_4^*$ as column sum of $\tilde{\Pi}_{3,22}$ is same as $\Pi_{3,22}^*$. For the other three:

$$\tilde{s}_3(i) = \tilde{\Pi}_{3,11}(i,i) = \sum_{k=1}^m \Pi_{3,11}^*(i,k) + \sum_{k=1}^n \Pi_{3,21}^*(k,i)$$

$$\tilde{s}_2(i) = \tilde{\Pi}_{3,22}(i,i) = \sum_{k=1}^n \Pi_{3,22}^*(k,i)$$

and hence by construction:

$$\|\tilde{s}_2\|_1 = \mathbf{1}^\top \Pi_{3,22}^* \mathbf{1} = \|s_2^*\|_1 - \mathbf{1}^\top \Pi_{3,21}^* \mathbf{1}.$$

$$\|\tilde{s}_3\|_1 = \mathbf{1}^\top \Pi_{3,11}^* \mathbf{1} + \mathbf{1}^\top \Pi_{3,21}^* \mathbf{1} = \|s_3^*\|_1$$

And also by our construction, $\tilde{s}_1 = s_1^* + c$ where $c = (\Pi_{3,21}^*)^\top \mathbf{1}$. As a consequence we have $\|c\|_1 = \mathbf{1}^\top \Pi_{3,21}^* \mathbf{1}$. Then it follows:

$$\sum_{i=1}^4 \|\tilde{s}_i\|_1 = \|s_1^* + c\| + \|s_2^*\|_1 - \mathbf{1}^\top \Pi_{3,21}^* \mathbf{1} + \|s_3^*\|_1 + \|s_4^*\|_1$$

$$\leq \sum_{i=1}^4 \|s_i^*\|_1 + \|c\|_1 - \mathbf{1}^\top \Pi_{3,21}^* \mathbf{1}$$

$$= \sum_{i=1}^4 \|s_i^*\|_1$$

So the objective value is overall reduced. This contradicts the optimality of $\Pi_3^*$ which completes the proof. □

## C  PROOF OF THEOREM 2.1

*Proof.* The proof is immediate from the Formulation 1. Recall that the Formulation 1 can restructured as:

$$ROBOT(\tilde{\mu}, \nu) = \inf_{P} \{OT(P, \nu) + \lambda \|P - \tilde{\mu}\|_{TV}\} .$$

where the infimum is taking over all measure dominated by some common measure $\sigma$ (with respect to which $\mu, \mu_c, \nu$ are dominated). Hence,

$$ROBOT(\tilde{\mu}, \nu) \leq OT(P, \nu) + \lambda \|P - \tilde{\mu}\|_{TV}$$

for any particular choice of $P$. Taking $P = \mu$ we get that

$$ROBOT(\tilde{\mu}, \nu) \leq OT(\mu, \nu) + \lambda \|\mu - \tilde{\mu}\|_{TV} = OT(\mu, \nu)) + \lambda \epsilon \|\mu - \mu_c\|_{TV}$$

Taking $P = \nu$ we get $ROBOT(\tilde{\mu}, \nu) \leq \lambda \|\nu - \tilde{\mu}\|_{TV}$ and finally taking $P = \tilde{\mu}$ we get $ROBOT(\tilde{\mu}, \nu) \leq OT(\tilde{\mu}, \nu)$. This completes the proof. □

