# OpenReview forum: "Outlier Robust Optimal Transport"
_ICLR.cc/2021/Conference — Reject_

### Official Review · AnonReviewer4 · 2020-10-25
**Good paper, as an advocate of optimal transport with truncated cost for robust estimation of distributions in data science**

**Rating:** 7
**Confidence:** 5

**Review:**

My evaluation:
* The main methodological contribution, claimed in the paper, is the equivalence between Formulation 1, which is an optimal transport (OT) problem regularized with the TV norm, and Formulation 2, which is a pure OT problem, with a truncated cost.
This equivalence is actually straightforward for people working with convex relaxations of nonconvex problems, more precisely linear programming (LP) relaxations. In fact, as soon as I read eq. 2.1, I wondered: why don't they consider instead the problem under the form, which appears later as eq. 2.3?! This equivalence is used for instance in the IEEE PAMI paper "What Is Optimized in Convex Relaxations for Multilabel Problems: Connecting Discrete and Continuously Inspired MAP Inference", 2014. Indeed, the truncated l1 cost is used with the same motivation as yours in convex relaxations of assignment problems in computer vision, for instance for depth map reconstruction in stereovision.
A short proof of the equivalence can be derived based on duality arguments: since the functionals are 1-homogeneous, in the dual domain we have an intersection of constrains, or equivalently the l_infinity norm which appears. It is known that the TV distance corresponds to OT with the 0-1 cost. So, since the regularized OT formulation is nothing but the infimal convolution of two convex functions, in the dual domain the constraints add up and we have the intersection of the two sets of constraints, which yields the minimum C_lambda of the two costs in the primal domain.
* Even if the methodological contribution is not a real one for experts in a specific area of optimization, in the context of robust methods for data science, the equivalence is interesting to present. So, the real contribution, in my opinion, is the application of this model to robust estimation, which in itself is sufficient for publication to ICLR. You should shorten the discussion about the equivalence and put it in the Appendix altogether.
* The application and the experiments show the relevance of the approach and its efficiency.

More detailed comments:
1) The regularized formulation 2.1 and the constrained formulation just before are completely equivalent: for every epsilon there exists lambda, and conversely, such that the solutions are the same (with ~mu = mu+s). So, if the constrained formulation "cannot distinguish between clean distributions", your formulation 2.1 has the exact same property.
2) "The ε-contamination model imposes a cap..." to have epsilon and not 2.epsilon, you should first define the total variation norm as ||s||_TV = (1/2)|s|(R^d) (because the 1/2 factor is not standard). Then remove "and ||s||_TV denotes..." after 2.2.
3) The Wasserstein-1 distance is in some sense robust to outliers. For instance, in 1-D, if one minimizes the OT with W1 cost between diracs at locations s_1,...,s_N, and searches over Theta = set of Diracs with amplitude N, the solution is one dirac at the median of the s_n. It is robust to outliers: changing one s_n to a very large or very small value does not change the median. But I agree that the truncated l1 cost is even more robust.

---

> ### Author Response · Authors · 2020-11-21
> **Response to Reviewer 4**
>
> Thanks for the feedback and suggestions for improvement. We have moved the definition of TV norm to earlier in the text as you suggested. Please see answers to the other comments below.
>
> **Proof of equivalence between formulations 1 and 2.**
>
> We thank the reviewer for outlining another proof of the equivalence, and we will elaborate on the duality-based argument in a revised version of the paper. However, we wish to point out that one of the insights from the more elementary proof is how to go between the optimal solutions of formulations 1 and 2.
>
> **If the constrained formulation "cannot distinguish between clean distributions", your formulation 2.1 has the exact same property.**
>
> The equivalence between the constrained and Lagrangian formulations depend on the two distributions in the arguments of the Wasserstein distance. In other words, as we vary the distributions (keeping $\epsilon$ fixed), the equivalent $\lambda$ will change. In our formulation, we are fixing $\lambda$ and varying the distributions, so the "solution paths" are different.
>
> **Truncated l1 cost is even more robust**
>
> The difference between the truncated l1 cost and the l1 cost is the difference between a loss function with bounded influence and a re-descending loss function (whose influence vanishes for extreme outlier values). The difference between the two types of loss functions is that extreme outliers will still affect the results with bounded influence loss functions (although their effects will be bounded), but they will not affect the results at all with re-descending loss functions. We are seeking this more stringent form of robustness in our work.

---

> > ### Comment · AnonReviewer4 · 2020-11-24
> > **Opinion on the authors' response**
> >
> > I thank the authors for their clear response to my points. After reading the other reviews, I still consider the paper as a good one and I will keep my score. I agree that the presence of concurrent papers is a good thing and I don't think that one should stress the relationship with unbalanced transport too much, since the motivations are rather different.

---

### Official Review · AnonReviewer3 · 2020-10-28
**Outlier robust OT is particular case of Unbalanced OT.**

**Rating:** 5
**Confidence:** 5

**Review:**

This paper proposes a modification of optimal transport to make it more robust with respect to outliers.

The basic idea is to truncate the cost used in the OT cost. The authors show the equivalence of this OT model and a TV regularization of OT (with a cost which coincides up to truncation).

Novelty is rather weak, the first formulation can be found as a special case of « Scaling algorithms for unbalanced optimal transport problems », Chizat et al (not mentioned in the paper). The model is already presented in the EMD literature by Peele and Werman (as mentioned in the paper).

Bibliography/background is insufficiently discussed.
Formulation 2.1 is essentially similar to that in « Generalized Wasserstein distance and its application to transport equations with source », by Piccoli and Rossi (Eq. 3).
The model shares also important similarities with partial optimal transport. Although the authors do discuss a link with Unbalanced OT, they mention relaxing the marginal constraint with Kullback-Leibler divergence, instead there is a closer link to partial OT.

Theoretical contribution: The equivalence between the two models is, to the best of my knowledge, new.

Experiments: there are two setups in which the model is used. First one on a mixture of gaussian distributions, a toy experiment comparing standard OT with truncated OT and shows as expected better performance in estimating the mean of the first « clean » distribution. Second experiment has the goal of identifying outliers and experiment it with GAN.
For this second application, a comparison with simple methods of outlier detection would have been welcome.

Writing: the paper is clearly written.

Valuable improvements for this work would focus on the experiment section. Comparing with other methods such as unbalanced OT and comparing with other methods of outlier detection.

---

> ### Author Response · Authors · 2020-11-21
> **Response to Reviewer 3**
>
> Thanks for the suggestions for improvement and additional references. Please see our general response for the summary of key changes and discussion of the relation to UOT (and partial OT). We emphasize that we do not claim mathematical novelty of Formulation 1 or Formulation 2 (as you noted, they both have appeared in prior works). Our key contributions are the application to outlier-robustness and the equivalence result. The practical implication of the equivalence result is that it is sufficient to use truncated-cost OT (which is easy to work with) when dealing with outliers, rather than handling more tedious UOT (or related) optimization problems.
>
> **Valuable improvements for this work would focus on the experiment section. Comparing with other methods such as unbalanced OT and comparing with other methods of outlier detection.**
>
> We have added comparison to UOT in Table 1 - the performance is quite similar as expected, but the optimization with truncated cost is simpler and scales better. Applying UOT in the outlier detection experiment is computationally challenging. We have updated the discussion in the experiment sections 4.1 and 4.2.
>
> For the outlier detection experiment, we added comparison to a distance thresholding approach in the end of Section 4.2 (based on the Reviewer 2 suggestion). Accuracy of this baseline is 85.4\%, inferior to the ROBOT accuracy of 90\%.

---

### Official Review · AnonReviewer2 · 2020-10-28
**Outlier Robust Optimal Transport**

**Rating:** 6
**Confidence:** 4

**Review:**

The authors propose to address the robustness over outliers for optimal transport (OT). They propose a new formulation based on penalizing the contaminated probability measures by a signed measure (which shares a close relation with unbalanced OT). The authors further derive an equivalent formulation by adjusting the cost matrix for the corresponding standard OT. Empirically, the authors evaluate their proposed approach on a toy example of robust mean estimation and outlier detection for data collection.

The idea to address the robustness over outliers for optimal transport is interesting. Although I have not checked the proof in detail, I think that the derived equivalence formulation (Formulation 2) which is a standard OT with the clipped cost. However, in my opinion, the problem (robustness to outliers) for OT is closely related to partial OT (and/or unbalanced OT) where one only optimal the partial alignment for probability measures (or relaxing the marginal constraints during optimization by divergence). (See [1])
+ Indeed, Formulation 1 shares a close relationship with the entropy transport problem (in Liero et al. [1]) where the divergence is a total variation. (See also [2])
+ There is a parallel work that appears in NeurIPS'2020 [3]. In [3], the authors also address the robustness of OT over outliers relying on unbalanced OT, and applies it into generative modeling, and domain adaptation.

It seems that the authors identify outliers from their distances to supports of the main distributions (which may explain the truncated cost in Formulation 2). Is it possible to just simply use a threshold to detect "outliers" as in applications in 4.2?

As in Algorithm 1 (and Figure 1), it seems that we can simply discard the constants $\Pi_{11}$ and $\Pi_{21}$ to reduce $\Pi$ into a matrix (n+m) x m

Some of my other concerns are as follow:
+ Although the new formulations are interesting, the authors should compare their approach with the partial OT and/or unbalanced OT which addresses the same concern for OT problem.
+ The assumption about a "clean" distribution for 1 of the 2 input ones for ROBOT is quite strong in applications. In this sense, I think that the unbalanced OT (as in [3]) may be more advantageous.

References:

[1] Matthias Liero, Alexander Mielke, and GiuseppeSavaré. Optimal entropy-transport problems and a new hellinger–kantorovich distance between positive measures. Inventiones mathematicae,211(3):969–1117, 2018.)

[2] Benedetto Piccoli and Francesco Rossi. Generalized wasserstein distance and its application to transport equations with source. Archive for Rational Mechanics and Analysis, 211(1):335–358,2014.)

[3] Yogesh Balaji, Rama Chellappa, Soheil Feizi. Robust Optimal Transport with Applications in Generative Modeling and Domain Adaptation. NeurIPS, 2020.

---

> ### Author Response · Authors · 2020-11-21
> **Response to Reviewer 2**
>
> Thanks for the questions, suggestions for improvement and additional references. Please see our general response for a summary of key changes and discussion of the relation to UOT (and partial OT). We emphasize that we do not claim mathematical novelty of Formulation 1 (as you noted, it is similar to UOT and related problems in the prior work). Our key contributions are the application to outlier-robustness and the equivalence to the simpler truncated-cost formulation. We answer your other questions below.
>
> **Is it possible to just simply use a threshold to detect "outliers" as in applications in 4.2?**
>
> We added comparison to a distance thresholding baseline in the end of Section 4.2. Accuracy of this baseline is 85.4\%, inferior to the ROBOT accuracy of 90\%. MNIST and Fashion MNIST images are not easily separable using Euclidean geometry, so this heuristic approach mistakes majority of the outliers for clean data.
>
> **As in Algorithm 1 (and Figure 1), it seems that we can simply discard the constants $\Pi_{11}$ and $\Pi_{21}$ to reduce $\Pi$ into a matrix (n+m) x m**
>
> You are correct. Those blocks are always 0 and do not affect the algorithm. They are there to maintain OT-like structure of Formulation 1 (discrete).
>
> **Although the new formulations are interesting, the authors should compare their approach with the partial OT and/or unbalanced OT which addresses the same concern for OT problem. The assumption about a "clean" distribution for 1 of the 2 input ones for ROBOT is quite strong in applications. In this sense, I think that the unbalanced OT may be more advantageous.**
>
> We added an equivalent two-sided version of Formulation 1, i.e. Formulation 3, in section 2.2, eq. (2.7) and extended the equivalence proof in Appendix A.2. Equivalence to Formulation 3 shows that the truncated-cost OT is an appropriate tool when both input distributions are corrupted. It also shows a very tight relation between ROBOT and UOT, and we added comparison to UOT in Table 1 verifying that the performance is similar. The key practical implication of our results is that it is sufficient to consider OT with truncated cost when handling outliers, rather than dealing with the more complicated UOT optimization. Please see general response for a discussion and summary of the relevant updates to the paper.

---

> > ### Comment · AnonReviewer2 · 2020-11-24
> > **UOT with KL divergence is as simple as Sinkhorn**
> >
> > Thank you for your clarification.
> >
> > >> Is it possible to just simply use a threshold to detect "outliers" as in applications in 4.2?
> >
> > >> We added comparison to a distance thresholding baseline in the end of Section 4.2. Accuracy of this baseline is 85.4%, inferior to the ROBOT accuracy of 90%. MNIST and Fashion MNIST images are not easily separable using Euclidean geometry, so this heuristic approach mistakes majority of the outliers for clean data.
> >
> > ---> So, it seems that using a threshold to detect outliers is OK for the experiments. In addition, the authors identify outliers based on their distances to supports of the main distributions (i.e., as in the submission, distances from outliers to supports of the main distributions are the same Euclidean? --- the ground cost). In my opinion, it seems that both methods are strongly affected by the threshold.
> >
> > >> The key practical implication of our results is that it is sufficient to consider OT with truncated cost when handling outliers, rather than dealing with the more complicated UOT optimization. Please see general response for a discussion and summary of the relevant updates to the paper.
> >
> > ---> For the truncated cost, it seems that this approach may be strongly affected by the threshold? It also requires a "clean" distribution which limits its application in practice. On the other hand, UOT with KL divergence is as simple as Sinkhorn for the balanced case (See algorithm in [1], and its complexity analysis in [2]). Moreover, UOT does not require any clean distribution assumption. It may also work when outliers exist in both considered distributions.
> >
> > References:
> >
> > [1] Frogner, C., Zhang, C., Mobahi, H., Araya, M., Poggio, T.A.. Learning with a Wasserstein loss. NIPS, 2015.
> >
> > [2] Khiem Pham, Khang Le, Nhat Ho, Tung Pham, Hung Bui. On Unbalanced Optimal Transport: An Analysis of Sinkhorn Algorithm. ICML, 2020.

---

> > > ### Author Response · Authors · 2020-11-24
> > > **We do not assume a clean distribution. UOT is implemented using Sinkhorn in the experiments.**
> > >
> > > > So, it seems that using a threshold to detect outliers is OK for the experiments. In addition, the authors identify outliers based on their distances to supports of the main distributions (i.e., as in the submission, distances from outliers to supports of the main distributions are the same Euclidean? --- the ground cost). In my opinion, it seems that both methods are strongly affected by the threshold.
> > >
> > > Performance of ROBOT is **clearly superior** to the threshold baseline. ROBOT is 90\% accurate and the threshold baseline is 85.4\% accurate.
> > >
> > > Both use Euclidean distance, but ROBOT relies on optimal transport rather than simple distance thresholding. To reiterate, to identify outliers with ROBOT, we solve an OT problem with truncated cost (Formulation 2) and then we recover Formulation 1 solution using Algorithm 2 to obtain indices of outliers. We also present empirical study of sensitivity of our method to the $\lambda$ choice in Figure 2. In Fig 2(a) we see that a wide range of $\lambda$ works well regardless of the outlier proportion. Fig 2(b) shows that choosing $\lambda$ is harder when outliers are similar to the clean data, but in this setting they are less detrimental.
> > >
> > > > For the truncated cost, it seems that this approach may be strongly affected by the threshold? It also requires a "clean" distribution which limits its application in practice. On the other hand, UOT with KL divergence is as simple as Sinkhorn for the balanced case.
> > >
> > > As mentioned above, we present an empirical study of the ROBOT sensitivity to the threshold $\lambda$ in Figure 2: it is **not** strongly affected by this choice.
> > >
> > > We **do not assume a clean distribution**. To illustrate that, we added Formulation 3 (eq. 2.7) to the paper, which allows modification of **both** marginals. Just like UOT, ROBOT is applicable when both input distributions have outliers. Please see last paragraph in our original response to your review for additional information.
> > >
> > > UOT also has a hyperparameter, the coefficient in front of the KL penalty. Formulation 3 is essentially a UOT with TV divergence, and $\lambda$ appearing there is the same as $\lambda$ in the truncated cost Formulation 2 as shown by our equivalence result (see Appendix A.2 for the extension of the equivalence result to Formulation 3). If you consider truncated cost formulation sensitive to the threshold, exactly the same reasoning applies to UOT with TV divergence, because it is the same problem with the same hyperparameter.
> > >
> > > We are aware of the Sinkhorn algorithm for UOT. That is how it was implemented in our experiments (see Section 4.1). It is also mentioned in the related work discussion in Section 2.1: "Chizat et al. (2018) proposed a Sinkhorn-like algorithm for entropy regularized UOT, but it is not amenable to stochastic optimization." As we explain there, the Sinkhorn algorithm is not as scalable as the stochastic algorithms for entropy-regularized OT proposed by Genevay et al. (2016). A stochastic algorithm for UOT was proposed only recently in the concurrent work of Balaji et al. (2020), but it requires two additional neural networks (total of four including dual potentials) to parameterize modified marginal distributions. Please see related work discussion in Section 2.1.

---

> > > > ### Comment · AnonReviewer2 · 2020-11-24
> > > > **Thanks for the clarification**
> > > >
> > > > Thank you for your clarification.
> > > >
> > > > I think the authors have had some improvements in the revised version, e.g., relation with UOT, no assumption about clean distribution anymore.
> > > >
> > > > I am happy to increase my rating (5 --> 6).

---

### Official Review · AnonReviewer1 · 2020-10-29
**Lack of theoretical grounding**

**Rating:** 4
**Confidence:** 4

**Review:**

SUMMARY
#######

The present paper proposes a way to robustify Optimal Transport (OT) with respect to outliers.

Assuming that one of the distributions on which OT is computed is $\epsilon$ corrupted (the second distribution being a parametrized distribution one wants to make close to the first one), authors propose to solve Kantorovich's problem for all distributions that are within an $\epsilon$-TV distance from distribution 1.

This problem is however hard to compute in practice, and an equivalent problem is proposed, based on a truncated cost.

It is formally proved that solving the second formulation gives a solution to the first one, and how to compute the optimal coupling matrix (in the discrete case).

Experiments are proposed, both on robust mean estimation for simulated data, and outlier detection on MNIST.



OPINION
#######

As for positive aspects, I find that:
- the paper is globally clear and well written, despite some minor clarity flaws (see below)
- the intuitions are well exposed and easy to follow


However, I find this contribution insufficient with respect to the following points:

- my main concern is about the lack of theoretical grounding for the proposed contribution. In particular:
a) other works with a different approach (see point on related works below) have derived consistency and convergence results for their estimator in the presence of outliers, does something similar hold for ROBOT?
b) formulation 1 uses explicitly the proportion of outliers $\epsilon$. What happens if the latter is only approximately known (which is much more likely in practice)?
c) formulation 2 uses an extra hyperparameter $\lambda$, related to $\epsilon$ in a very complex way that is hard to interpret, and for which no selection procedure is proposed except cross validation.
d) in the end, the proposed method thus boils down to the introduction of a new threshold parameter $\lambda$, and a "test all possible values, one should yield a better result" strategy. I find it a bit disappointing not to have more theoretical insights. I agree however that the threshold makes perfect sense here, and that ROBOT is "always" better than vanilla OT in the proposed experiments. But this behavior is not that uncommon for threshold parameters ($epsilon$-insensitive, huber loss) and it is difficult to say something else than "we have added another hyperparameter".

- p.1 "can have an outsized impact": what if the cost function is already robust (e.g. Wasserstein 1)? Have authors noticed differences with respect to the cost used?

- p.1 "there are no methods in the literature for achieving outlier-robustness with MKE": the following two references might be relevant
a) Staerman, Guillaume, et al. "When OT meets MoM: Robust estimation of Wasserstein Distance." arXiv preprint arXiv:2006.10325 (2020).
b) Balaji, Yogesh, Rama Chellappa, and Soheil Feizi. "Robust Optimal Transport with Applications in Generative Modeling and Domain Adaptation." arXiv preprint arXiv:2010.05862 (2020).

- p.2 "the value of outliers is arbitrary": the standard framework of OT is on bounded observations. In that case, how large can be the impact of bounded outliers?

- p.5 1 algorithm + 1 explicative paragraph + 1 figure seems a bit too much for a procedure which is not that complex

- from what I have seen in the core text and proofs, only getting a solution to Robot_1 from a solution to Robot_2 is proposed, while Thm 3.1 suggests both directions are possible

- p.12 what does "suppose" mean? I might have missed something, but you cannot suppose anything here. Or the contradiction you get in the end might refute this assumption, rather than the fact that $\Pi_2^*$ is optimal.

- What happens if $\nu$ is also corrupted?



MINOR COMMENTS
##############

p.1 $\mu$ and $\nu$ instead of $P_1$ and $P_2$ in eq. (1.1)
p.1 $c$ is not defined in eq. (1.1)
p.1 $\nu_\theta$ is not defined in eq. (1.2) (although it is globally understandable)
p.2 $\epsilon$ should be in the interval [0, 1/2]?
p.2 and after TV *distance* and not *norm*
p.2 TV subscripts in the last paragraph
p.3 C\lambda*(x,y)* in eq. (2.3)
p.3 to formulate *a* discrete
p.3 *a* discrete analog of
p.3 $\Delta^{m-1}$ is not defined
p.3 it should be better explained why one needs to consider the augmented versions
p.4 $s_1$ and $t_1$ are not defined. Maybe $s$ and $t$ is enough, since there is not $s_2$
p.4 shouldn't it be + 2 *\lambda in eq. (2.4)?
p.4 $1_m$ is not defined
p.4 it is a bit misleading to have the same notation $\mu_n$ for the vectors and the distributions
p.4 "we can recover optimal solution" --> "optimal coupling" may be more clear, as the solutions are supposed to be the same
p.5 the block matrix notation of Sec. 3.2 is not very standard, and could be replaced by the one used in Alg. 1
p.5 we *are* not moving
p.6 in ou*r* second experiment
p.11 this is not an*d* optimal solution
p.12 *&* symbol, quite unusual



OVERALL EVALUATION
##################

Although the equivalence result is interesting, I find the contribution slightly insufficient to warrant acceptance, as the proposed method essentially boils down to adding an extra threshold hyperparameter, without more theoretical discussion.


--- EDIT POST REBUTTAL ---

I thank the authors for their answer and their efforts in editing the submission. I have also read other reviews and replies. However, my stance on the paper did not really change as I find the contribution insufficient for acceptance.

PS: when I wrote "Formulation 1 uses explicitly the proportion of outliers", I was referring to the display equation above eq. (2.1). I did not realize the term "formulation" was already formally used in the paper to refer to another equation.

---

> ### Author Response · Authors · 2020-11-21
> **Response to Reviewer 1 [Part 1]**
>
> Thanks for the questions and suggestions for improvement. Please see our general response for a summary of key changes and comments regarding other papers studying outlier-robustness in OT. We answer your questions below.
>
> **Other works with a different approach (see point on related works below) have derived consistency and convergence results for their estimator in the presence of outliers, does something similar hold for ROBOT?**
>
> We have added Theorem 2.1 to the paper to strengthen theoretical guarantees as you suggested, but before discussing it, we would like to reiterate that the related works you mentioned should not be considered as prior art according to the ICLR 2021 Reviewer guide. Balaji et al. (2020) became publicly available **after** the ICLR submission deadline and Staerman et al. (2020) is not published in peer-reviewed conference proceedings or journals.
>
> Balaji et al. (2020) is the more similar work, since they also draw inspiration from UOT. They do not have a consistency result. Their theoretical statement justifying robustness is Theorem 2, which establishes an upper bound on their modification of OT. Our Theorem 2.1 is a similar guarantee (it can be easily re-stated as a multiplicative bound by saying that there exists $k$ such that OT$(\mu,\nu) = k\lambda$ and noting that TV distance is bounded by 1). Our statement has a slight advantage as it guarantees that an adversary can not increase the ROBOT distance arbitrarily. In their bound, an adversary can increase $k$ by modifying the contamination distribution.
>
> Staerman et al. (2020) consider a similar problem, but use a very different approach -- they modify Wasserstein-1 dual replacing expectation with a median-of-means (MoM). They show that the resulting estimator is consistent, but it relies on the assumption (Assumption 3) that the fraction of outliers vanishes in the large sample limit. Compared to the standard Huber's $\epsilon$-contamination model (our setting), this is a very restrictive assumption. With the help of Theorem 2.1 that we added to the paper, we can also establish consistency under their assumptions. We are happy to provide the details if you request.
>
> **Formulation 1 uses explicitly the proportion of outliers $\epsilon$. What happens if the latter is only approximately known (which is much more likely in practice)? Formulation 2 uses an extra hyperparameter $\lambda$, related to $\epsilon$ in a very complex way that is hard to interpret, and for which no selection procedure is proposed except cross validation.**
>
> Neither Formulation 1 nor Formulation 2 has $\epsilon$ in it. The only hyperparameter we have is $\lambda$ and we do propose a heuristic for selecting it. Please see Section 4.2. Moreover, $\lambda$ is an interpretable hyperparameter as it is closely tied to the distance between samples. Our heuristic uses this interpretation suggesting to set $\lambda$ by subsampling from the clean data; no knowledge regarding outliers or their proportion is needed. We also present empirical study of sensitivity of our method to the $\lambda$ choice in Figure 2. In Fig 2(a) we see that a wide range of $\lambda$ works well regardless of the outlier proportion (which is rarely known in practice as you noted). Fig 2(b) shows that choosing $\lambda$ is harder when outliers are similar to the clean data, however in this setting they are less detrimental.
>
> **It is difficult to say something else than "we have added another hyperparameter"**
>
> We do not think this is a fair evaluation of our work. Many ML methods are based on hyperparameters, e.g. the celebrated LASSO. Hyperparameters can be useful or not, depending on their interpretability and their sensitivity. In our previous answer we have emphasized that hyperparameter in our method is interpretable, allowing us to propose meaningful selection heuristics, and our method is not overly sensitive to the hyperparameter.

---

> > ### Author Response · Authors · 2020-11-21
> > **Response to Reviewer 1 [Part 2]**
> >
> > **What if the cost function is already robust (e.g. Wasserstein 1)?**
> >
> > Although the Wasserstein-1 distance is "more robust" than Wasserstein-2, it is not robust in a mean versus median sense. To elucidate, it is enough to perturb a single support point of one of the input distributions to arbitrarily increase Wasserstein-1. On the contrary, ROBOT distance is always bounded above.
> >
> > **The standard framework of OT is on bounded observations. In that case, how large can be the impact of bounded outliers?**
> >
> > Boundedness of the domain is a common mathematical assumption made for convenience. Many theoretical results in OT are also established without this assumption. For bounded domains, impact of outliers can be bounded by the diameter of the domain.
> >
> > **From what I have seen in the core text and proofs, only getting a solution to Robot-1 from a solution to Robot-2 is proposed, while Thm 3.1 suggests both directions are possible. What does "suppose" mean?**
> >
> > Please see Appendix A.1 page 13 around "we create a feasible solution namely $\Pi^*_{2,new}$ of Formulation 2 as follows:" where we construct solution to F2 from F1. We replaced "suppose" in page 12 (now page 13) with "define" for clarity.
> >
> > **What happens if $\nu$ is also corrupted?**
> >
> > Using truncated cost continues to be an appropriate solution path. We added an equivalent two-sided version of Formulation 1, i.e. Formulation 3, in section 2.2, eq. (2.7) and extended the equivalence proof in Appendix A.2. Please see the corresponding paragraph in the draft and the general response for a discussion.
> >
> > **Minor comments**
> >
> > We have fixed the typos/ambiguities in the notations.

---

### Author Response · Authors · 2020-11-21
**Summary of the key changes and related work discussion**

We thank the reviewers for their feedback. We are glad all reviewers agree that unbalanced OT (UOT) is a meaningful way of promoting outlier robustness for OT-related problems in ML. We are also thankful for the additional references to prior works studying UOT.

Our Formulation 1 (F1) is indeed strongly related to UOT. To further strengthen this connection we added another equivalent formulation: a two-sided version of F1 in Formulation 3 (eq. 2.7, see Appendix A.2 for equivalence proof). We are not claiming that Formulation 1 is novel mathematically. Rather, our contribution is to notice its applicability to promoting outlier robustness in MKE and related problems, and, crucially, to establish its equivalence to a simpler, convenient for optimization, Formulation 2 (F2). At a first glance, F2 appears over-simplistic and unsuitable for robustness, but our proof of equivalence justifies its usage in practice. As several reviewers have noted, the equivalence result is novel.

We note that **at the time of submission, there was no prior work considering UOT for outlier-robustness in ML.** Reviewers 1 and 2 reference work of Balaji et al. (2020) considering UOT for robustness. Their paper became publicly available on **October 12th, after the submission deadline.** It was  impossible for us to know about this paper while we were writing ours. Please also see ICLR 2021 Reviewer guide FAQ regarding citing and comparing with very recent work. We ask that Reviewers and Area Chair take this into account in their final evaluation. That said, we now cite Balaji et al. (2020) as concurrent work.

**Below we summarize key changes to the paper:**

* In place of the Remark regarding UOT, we added a Related work paragraph in Section 2.1 discussing UOT and partial OT work, including Balaji et al. (2020). We also discuss the concurrent work of Staerman et al. (2020) suggested by Reviewer 1, which tackles robustness in OT via a median-of-means based modification to the Wasserstein-1 dual rather than a UOT-based approach. We were unaware of this work previously: the first version of this paper appeared on ArXiv during the summer, but it has not been published. According to the ICLR 2021 Reviewer guide: "they [authors] may be excused for not knowing about papers not published in peer-reviewed conference proceedings or journals." We also note that the presence of two concurrent works studying the same problem suggests relevance of the topic to the community. The key take-away of our prior work discussion is that all prior and concurrent works, including our Formulation 1, are challenging to optimize. Our equivalence result shows that a simple OT with truncated cost can be used instead, which is easier to work with and combine with other OT optimization techniques in the literature.

* Inspired by the Reviewers' agreement regarding utility of UOT for robustness, we added Formulation 3 (eq. 2.7) to strengthen this connection. F1, F2 and F3 are all equivalent; please see Appendix A.2 for F1 to F3 equivalence proof. F3 and the extended equivalence proof demonstrate that even when both input distributions are corrupted with outliers, using truncated cost is an appropriate solution path (our equivalence proofs are also sufficient to recover solution to F3 from F2 when outlier detection is of interest, i.e. by recovering F1 solution from F2 and then F3 solution from F1).

* We added Theorem 2.1 in Section 2.1, strengthening the robustness justification of our formulations. It is an upper bound guaranteeing that the ROBOT distance is bounded regardless of the contamination distribution.

* We added comparison to a UOT-based GAN in Table 1. To solve UOT, we used the Sinkhorn-like algorithm of Chizat et al. (2018), available through the POT library. The performance is unsurprisingly similar, since we showed that OT with truncated cost is strongly related to UOT (i.e. F3). To reiterate, solving OT with truncated cost is much simpler and is easy to scale using existing stochastic optimization algorithms for entropy-regularized OT. In the outlier-detection experiment in Section 4.2, we added a comparison to a distance-based heuristic approach based on the Reviewer 2 suggestion.

---

### Decision · Program_Chairs · 2021-01-07
**Final Decision**

**Decision:**

Reject

**Comment:**

The paper proposes a novel approach to detect outliers using Optimal transport. the authors prove a very interesting relation between Outlier robust OT and solving OT with a  thresholded loss. Numerical experiments show that the proposed approach indeed work for outlier detection.

The paper had mixed reviews and the comments and changes from the authors were appreciated. The comments about recent (and contemporary) references were not taken into account in the final decision following ICLR guidelines.

One major concern that appeared during discussion was the fact that one important claimed contribution is the ability to perform outlier detection, the proposed method is never evaluated or compared to the numerous existing outlier detection methods. It works on a toy example and seem to provide a robust way to train a robust GAN but the experiments are very limited. Also the claim from the authors that the method scales are not really true. The proposed approach requires solving an exact OT of complexity O(N^3log(N)), while one can use an approximated entropic solver on the thresholded loss it does not solve the ROBOT problem anymore and the relations between the problem does not exist anymore in this case (or are more similar to UOT).

The concerns detailed above and the limited novelty of the contributions (most of the formulations proposed in the paper are already existing in the literature) suggest that the paper in its current iteration  is too borderline for being accepted in a selective venue such as ICLR. The method and the relations uncovered are interesting and the AC encourages the authors to continue work on the proposed method and provide more detailed experiments illustrating and comparing the method to baselines for outlier detection.